# Natural and anthropogenic contributions to the hurricane drought of the 1970s–1980s

Raphaël Rousseau-Rizzi [1] & Kerry Emanuel [1]

Atlantic hurricane activity experienced a pronounced lull during the 1970s and 1980s. The current explanation that anthropogenic aerosol radiative forcing cooled the sea surface locally fails to capture the magnitude of this large decrease in activity. To explain this hurricane drought, we propose that the radiative effects of sulfate aerosols from Europe and North-America decreased precipitation in the Sahara-Sahel region, leading to an enhancement of dust regional emissions and transport over the Atlantic. This dust in turn enhanced the local decrease of sea-surface temperature and of hurricane activity. Here, we show that dust emissions from the Sahara peaked in phase with regional sulfate aerosol optical thickness and Sahel drought conditions, and that dust optical depth variations alone can explain nearly half of the sea-surface temperature depression in the 1970s and 1980s.

North-Atlantic hurricane activity, as measured by the power dissipation index[1] or the accumulated cyclone energy[2], was much smaller during the 1970s and 1980s than in the preceding or following decades[3–5]. In a changing climate, the high hurricane activity, and destructiveness we have witnessed in the recent decades[6] pose the pressing question of whether the hurricane drought of the 1970s and 1980s was natural or anthropogenic in nature. In the former case, we might expect similar droughts to recur, providing a respite from hurricane hazards. In the latter case, we need to consider the recent hurricane activity as the new normal and prepare accordingly[7].

The hurricane power dissipation index depends on the number of hurricanes during each season, their intensity, and their duration. These in turn depend on a variety of proximal environmental influences like variations in mid-tropospheric saturation deficit, vertical wind shear, and potential intensity, a theoretical upper limit on tropical cyclone (TC) intensity imposed by the thermodynamic environment[8,9]. Potential intensity theory predicts stronger tropical cyclones when the sea surface is anomalously warm locally[10], and indeed, North Atlantic power dissipation is highly correlated with summertime tropical Atlantic sea surface temperatures (SSTs)[1,11]. This is especially true at long timescales, and SST was depressed on average in the 1970s and 1980s, concurrently with the hurricane drought[1].

The evidence presented in a steadily growing number of papers using both statistical[12], and numerical[13–15] methods suggests an important influence of anthropogenic sulfate aerosol radiative effects on Atlantic ocean SSTs during the hurricane drought. Indeed,

emissions of $SO_2$, a sulfate aerosol precursor, increased rapidly to peak in Europe and North America in the 1970s and 1980s, before decreasing rapidly due to emission controls[16]. However, despite accounting for sulfate radiative forcing, climate model simulations fail to represent the magnitude of SST variations at multidecadal time scales, in the tropical North-Atlantic and the main development region[17–20]. As a result, models also fail to capture the full observed depression of hurricane activity of the 1970s and 1980s[14,15]. This contrast between a large statistical dependence of TC activity on the emissions of $SO_2$[12], and a relatively small modeled dependence suggests that the effects of anthropogenic sulfates could be amplified locally by positive climate feedbacks that are not well captured in climate models. We propose that Saharan dust emissions could provide such a feedback.

Dust lofted from the Sahara-Sahel gets transported in enormous amounts over the tropical North-Atlantic all the way to the Caribbean[21], where its concentration peaked in the 1970s and 1980s[22]. In that region, eolian dust has the largest radiative effect of any aerosol species present[23], which acts to depress SST[24,25] and reduce hurricane activity[26,27]. In addition, dust aerosol's mean and variance are vastly underrepresented in climate models, at least up through CMIP5[28]. This might explain why, thus far, GCM simulations fail to capture the observed SST depression of the 1970s and 1980s, unless dust optical thickness is prescribed (e.g., ref. 27). Yet, despite the need for a deeper understanding of the causes of dust variation, there is comparatively little literature on this topic.

[1]Lorenz Center, Massachusetts Institute of Technology, 77 Massachusetts Avenue, Cambridge, MA 02139, USA. ✉e-mail: rrizzi@mit.edu

In the present study, we propose that the high dust aerosol concentrations from the 1970s and 1980s did not result from internal variability, but occurred as feedback to anthropogenic sulfate radiative forcing. Further, we propose that the radiative effects of this dust feedback might have exceeded those of the sulfates themselves in the tropical North Atlantic main development region. An important caveat to this statement is that the radiative effects of the sulfate aerosols themselves are uncertain[29]. Perturbations to SST in the tropical North Atlantic are also thought to be enhanced by surface wind speed and cloud feedbacks, which may be related to the effects of dust[30,31].

At the peak of $SO_2$ emissions, in the 1970s and 1980s, sulfate aerosols originating from Europe were swept southward across the Mediterranean and over North Africa by dominant lower tropospheric winds[32], where they acted to cool the regional climate[33], which weakened the Saharan heat low and the West-African monsoon[34]. Simultaneously, sulfate aerosol forcing around the North-Atlantic basin, due mostly to emissions from Europe and North America, weakened the inter-hemispheric temperature gradient during summer[35], which may have reduced the northward extent of the African monsoon. Both mechanisms have been shown by multiple studies to have caused or worsened a concurrent hydrological drought in the Sahel[36–40]. Volcanic stratospheric aerosols are similarly hypothesized to modulate hydrological drought conditions in the Sahel, especially when they are asymmetric about the equator[41,42]. In turn, drought conditions in the Sahel correlate well with the dust that covers the Atlantic[43].

Hence we suggest that past hurricane activity has been mostly controlled by local dust forcing, the variability of which has been indirectly modulated by remote sulfate aerosol forcing. A paleoclimate analog to this hypothesis can be found in the case of the mid-Holocene green Sahara, which was due to orbital forcing changes[44] and associated with reduced dust emissions and transport[45], which would have enhanced tropical cyclone activity[46]. To help explain the 1970s–1980s TC drought, we first quantify the associated SST and power dissipation index variations. We then estimate dust cover over the tropical North-Atlantic during the 20th century and relate it to sulfate aerosol forcing. Finally, we show that dust direct radiative effects alone can explain a large fraction of tropical North-Atlantic SST variance at long time scales, and hence are necessary to explain the hurricane drought of the 1970s–1980s.

## Results

### Magnitude of the hurricane drought

First, we compare the tropical cyclone activity in the Atlantic during the 1970s–1980s to that during the period extending from 1960, when TC activity estimates become credible[3], to 2017. Activity during these two periods is illustrated in Fig. 1a, which shows that major hurricanes (hurricanes with lifetime maximum wind speeds in excess of 50 m s$^{-1}$) are responsible for almost all of the difference in power dissipation index between the two periods. During the hurricane drought, there were fewer storms of all categories, with the largest relative differences occurring for major hurricanes. On average, across all categories, the power dissipation index was about 55% larger in 1960–2017 than in 1970–1990. Separating PDI into its three components, storm number, intensity, and duration (see the "Methods" section), shows that the 55% change in PDI is due to a 22% change in hurricane number, a 14% change in intensity (proportional to wind speed cubed) and an 11% change in storm duration. In other words, during the 1970s and 1980s hurricane drought, storms were less numerous (the biggest contribution to PDI change), weaker and shorter-lived. On average during the hurricane drought, the main development region's summer SST anomaly was −0.13 K, which is substantial over such a long time. The main development region is defined here as the Tropical North-Atlantic region extending from 6 to 18N and from 60 to 20W.

The strong relation between summertime tropical Atlantic SSTs and the power dissipation index during that time is further illustrated

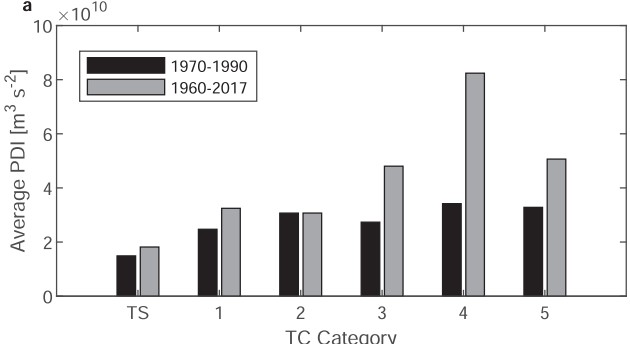

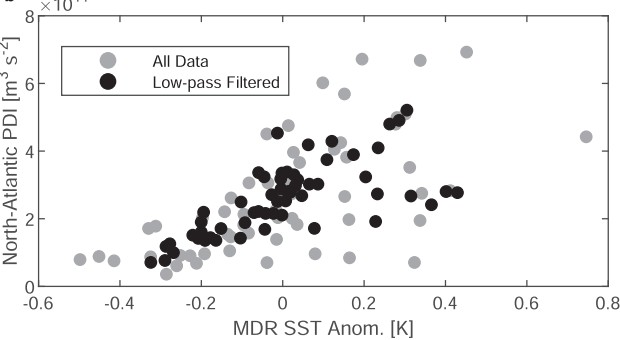

**Fig. 1 | Power dissipation index and sea-surface temperature anomaly.**
**a** Average power dissipation index per year and per storm category for the period from 1970 to 1990 (black bars) and from 1960 to 2017 (gray bars). **b** Scatter plot of power dissipation index against main development region sea surface temperature anomaly (gray, $R = 0.63$), and 7-year cutoff low-pass filtered power dissipation index against main development region sea surface temperature anomaly (black, $R = 0.68$), from 1960 to 2017.

in Fig. 1b which shows a scatterplot of the power dissipation index against the hurricane season (August–October) main development region SST anomaly during the 1960–2017 period described above. Hence, throughout this study, the main development region SST anomaly will be used as a predictor of Atlantic hurricane activity at decadal-to-multidecadal time scales, which is warranted by our physical understanding of TC energetics and by the strong correlations presented in Fig. 1.

### Dust reconstruction

Then, similarly to Mahowald et al.[47], we reconstruct dust variations by using Sahel precipitation as a proxy. In reality, rather than the Sahel hydrological drought itself, the closely associated large-scale low-level wind changes[48,49] are thought to have been the proximal cause of enhanced emission and transport of dust[50,51]. However, since precipitation has been directly observed for over two centuries in the Sahel[52], drought is a more reliable proxy for dust than reanalysis wind data for the first part of the 20th century.

Figure 2a shows the reconstructed dust optical depth (see the "Methods" section) over the main development region (blue), based on a Barbados dust record (black)[53], on a Sahel precipitation index (blue) averaged over the region [20-10N, 20W-10E][54], and on AVHRR satellite measurements of dust optical depth over the main development region during the peak of the 1980s[22]. The correlation between the low-pass filtered Barbados dust and the Sahel precipitation index is $R = -0.77$, which is very strong and justifies the use of the Sahel precipitation index as a proxy for dust. In our analysis, the low-pass filtered reconstructed dust optical depth varies between a minimum of 0.04 and a maximum of 0.29 (in the early 1980s) over the course of the 20th century. Confidence intervals on the dust reconstruction are also

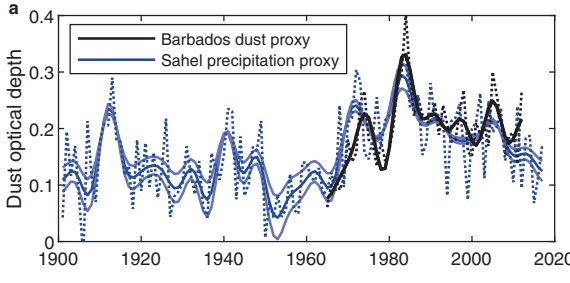

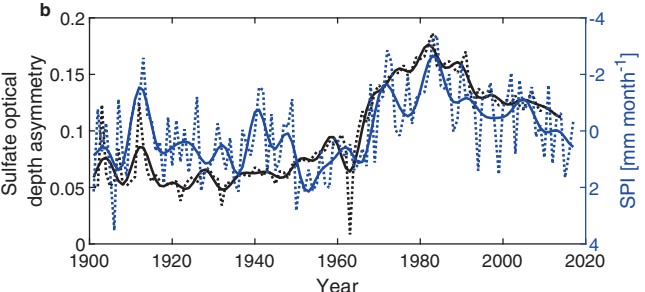

**Fig. 2 | Dust optical depth proxies and sulfate aerosol asymmetry. a** Barbados summer dust measurements rescaled by satellite measurements of main development region dust optical depth during the 1980s (black), main development region dust optical depth reconstruction based on the Sahel precipitation index (SPI) proxy (dark blue) and 95% confidence interval on the SPI proxy (pale blue). Dotted lines are not filtered, and full lines are low-pass filtered. **b** Low-pass filtered anthropogenic and volcanic sulfate aerosol optical depth asymmetry (black), and SPI (blue). The *y*-axis for the precipitation index is reversed so that peaks indicate dry years. Dotted lines are not filtered, and full lines are low-pass filtered.

shown (light blue), and on average, during the 1970s and 1980s, dust optical depth is $0.043 \pm 0.010$ higher than the average of the 1960–2017 period (see the "Methods" section).

Figure 2b shows the Sahel precipitation in mm/month (blue), along with a simulated index for the hemispheric asymmetry in sulfate aerosol optical depth (see the "Methods" section) of volcanic and anthropogenic origins (black). The precipitation timeseries is the same as in Fig. 2a, except it is not rescaled here. The asymmetry is due in large part to the change in anthropogenic emissions in the northern hemisphere since southern hemisphere emissions are comparatively small in the region considered, hence, this simple index can be considered to capture both the forcing effects on the Saharan heat low or on the interhemispheric temperature gradient. The correlation between this low-pass filtered sulfate aerosol asymmetry index and the Sahel precipitation index is $R = -0.76$, supporting the idea that European sulfate emissions enhanced Sahel drought.

The contribution of volcanoes to the asymmetry index, like that of anthropogenic emissions, depends on the location of the aerosols, and the eruptions that have the largest influence on Sahel precipitation are not always the strongest, like that of Pinatubo (1991), but the most asymmetric[41]. For example, the strongly asymmetric El Chichon eruption (1982) immediately precedes the most severe drought on record in the Sahel, and the dustiest year on record in Barbados. One other notable example is the eruption of Novarupta (1912) which was confined to the northern hemisphere and is also associated with a large spike in drought conditions (see Fig. 2b).

## Simulations of dust radiative impacts
Next, we attempt to estimate how much those dust variations might have influenced tropical North-Atlantic SSTs during the 1970s and 1980s. To do so, we test the sensitivity of SST to dust aerosol direct radiative forcing in single-column model (SCM) simulations, under a

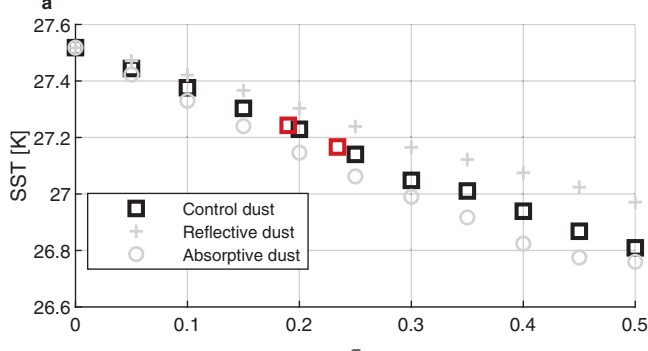

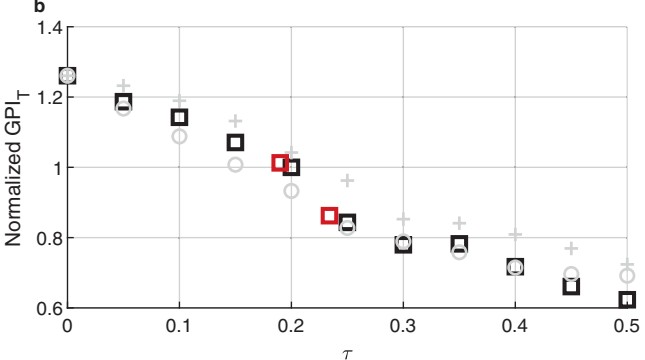

**Fig. 3 | Sensitivity of sea-surface temperature (SST) and the thermodynamic component of the genesis potential index (GPI$_T$) to dust optical depth in the single-column model simulation. a** Simulated SST and **b** GPI$_T$ as a function of dust optical thickness ($\tau$) at 0.55 μm, under a weak temperature gradient constraint. The red dots are the values of SST and GPI$_T$ for dust concentrations representative of the 1970s–1980s hurricane drought ($\tau = 0.23$) and of the entire period from 1960 to 2017 ($\tau = 0.19$). The gray profiles represent simulations with single scattering albedo ($\omega_0$) perturbed so that the dust is more reflective ($\omega_0 = 0.94$) or more absorptive ($\omega_0 = 0.84$). Similar simulations where the asymmetry parameter ($\hat{g}$) is perturbed to values of 0.58 and 0.78 yield similar results.

weak-temperature gradient constraint (see the "Methods" section). Figure 3a shows that the sensitivity of SST to shortwave optical depth ($\tau$) is linear and has a value $\delta SST/\delta\tau = -1.4\,K\,\tau^{-1}$ (black squares). This result means that the estimated dust optical perturbation of 0.043 during the 1970s and 1980s can explain a 0.06 K depression over those 20 years. Modifying dust optical properties such as the single scattering albedo or the asymmetry parameter to unlikely values yield sensitivities between $\delta SST/\delta\tau = -1.3\,K\,\tau^{-1}$ and $\delta SST/\delta\tau = -1.8\,K\,\tau^{-1}$ (gray symbols). Combining these bounds in dust optical properties with the uncertainty on dust optical depth yields bounds of $-0.04$ and $-0.10\,K$ on the 1970s and 1980s main development region temperature anomaly that can be explained by dust. Hence, dust can explain a substantial fraction of the observed $-0.13\,K$ anomaly during that period, but not all of it. The remainder may be due to direct forcing by sulfates (e.g., ref. [13]) and to wind and cloud feedbacks (e.g., ref. [55]). We note that certain uncertainties, like that on the satellite retrievals of dust optical depth[56] are hard to assess and were not accounted for in these bounds.

We next turn our attention to the sensitivity of TC activity-relevant physical parameters to dust aerosol optical depth. Figure 3b shows the variations, as a function of dust optical depth, of the thermodynamic component of the genesis potential index[57] defined as

$$GPI_T = \chi^{-4/3}(PI - 35)^2, \qquad (1)$$

where PI is potential intensity[8] and $\chi$ is mid-tropospheric saturation deficit[58]. The genesis potential index is proportional to the number of

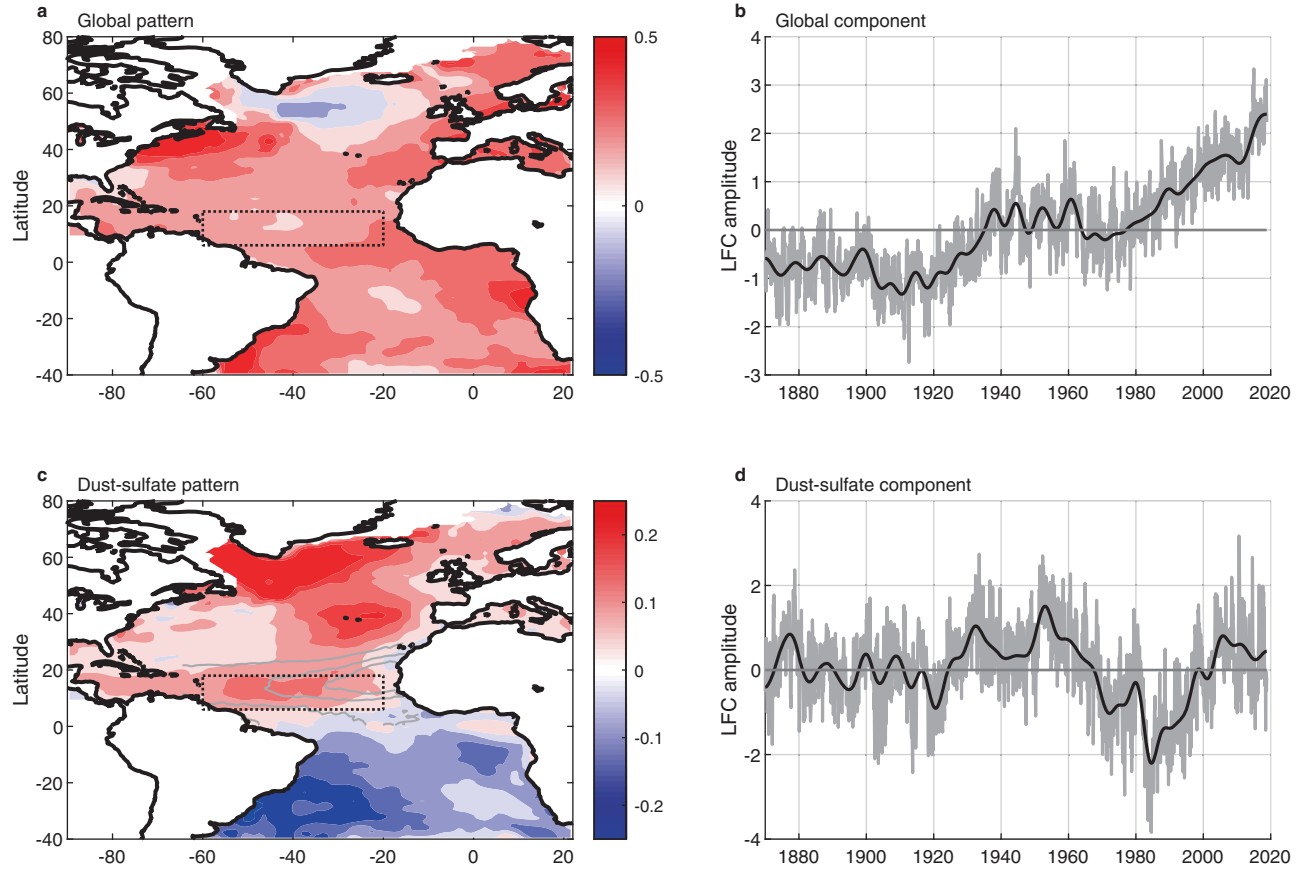

**Fig. 4 | Low-frequency component analysis global and multidecadal modes of sea-surface temperature (SST) variations.** Patterns of sea-surface temperature (color shading) associated with the global mode (**a**) and the dust-sulfate mode (**c**). The dust-sulfate pattern is plotted along with dust aerosol optical depth contours[22] for $\tau = [0.15\,0.3\,0.45]$ (gray contours), and the main development region is identified (dotted black box). Associated components for the global mode (**b**) and the dust-sulfate mode (**d**), including variability at all frequencies (gray) and only at low frequencies (black).

storms that form at a given location. As dust increases, PI decreases and $\chi$ increases, leading to a decrease in $GPI_T$. In Fig. 3b, $GPI_T$ is normalized by its value at $\tau = 0.2$, which is representative of the dust cover over the main development region. In Fig. 3b, the red squares correspond to two simulations based, respectively, on the high dust optical depth of the 1970s and 1980s, and on the average of the 1960–2017 period. This shows that dust variations may have resulted in a 15% decrease in the genesis potential index during the 1970s and 1980s. Considering that there were 22% fewer observed hurricanes during that period and that the effects of dust on wind shear were neglected in our computation, this shows that dust variations can explain a large fraction of the decrease in hurricane numbers in the 1970s and 1980s.

## Modes of SST variations

Finally, we seek objective modes of SST variability that can be associated with the sulfates and dust. To do so, we apply the low-frequency component analysis method[59] (see the "Methods" section) to SSTs over the entire Atlantic basin, from 1870 to 2017, and obtain 25 SST patterns and their associated components. Figure 4 shows the two low-frequency components and low-frequency patterns that have the highest low-frequency to total variance ratios and jointly represent most of the low-frequency variance over the whole basin.

Even though the analysis is restricted to the Atlantic basin, the first component (Fig. 4b) is very strongly correlated ($R = 0.98$) to low-pass filtered tropical mean SST, and hence will be called the global warming component. Since the components produced by the low-frequency component analysis method are uncorrelated from one another, this

high correlation means that none of the other components are representative of global changes. Correspondingly, the first pattern (Fig. 4a) will be called the global warming pattern.

The variations of SST associated with the global mode and averaged over the North-Atlantic main development region have a regression coefficient with the tropical mean SST of 0.97. This means that, in this mode, for every degree the tropics warm, the main development region warms by very nearly a degree, and hence allows us to treat the contribution of all the other modes as spatial anomalies with respect to the tropical average. This is a very useful result because TC activity-relevant parameters such as potential intensity or the genesis potential index are much more sensitive to local SST changes than to global ones (e.g., refs. 10, 60). Hence, over a timescale of a few decades, the global mode is unimportant to explain the North-Atlantic power dissipation index, relative to several higher order modes, and moving forward we will neglect it in our analysis.

Figure 4c shows the second low-frequency pattern, which exhibits large variance in a horseshoe pattern in the Northern Hemisphere starting from a maximum in the region of the subpolar gyre and decreasing progressively until the tropical North Atlantic. Measured dust optical depth contours are overlaid on the pattern[22]. The local interhemispheric temperature difference associated with this mode is a factor 3 larger than that associated with any other pattern. Considering the fact that preindustrial climate simulations do not exhibit large interhemispheric temperature difference variations[59], this result suggests that the second pattern is due to hemispherically asymmetric sulfate forcing. Further, the second low-frequency component (Fig. 4d) is the only component that correlates well with Sahel drought

($R = 0.69$). The fact that no other SST mode correlates well with Sahel drought suggests a relation between the second mode specifically and Sahel precipitation, or a common driver, and suggests that this mode is associated with dust emissions and radiative forcing. Finally, we note that the variance of this second mode from 1870 to 1950 is about half that from 1950 to 2018, which is coherent with the previous literature[61] and is consistent with a radiative forcing event in the second half of the 20th century. Henceforth this mode will be called the dust-sulfate mode.

Using Parseval's theorem (see the "Methods" section), we find that the dust-sulfate mode explains 88% of the local SST variance at time scales between 20 and 100 years in the main development region. Averaged over the 1970s and 1980s, the dust-sulfate mode is responsible for a −0.14 K change in SST averaged over the main development region. This is multiple times larger than the contribution by any other mode (except the global mode which is not relevant here). Averaged over the same period, the SST anomaly in the main development region is −0.13 K, which is very similar and consistent with the fact the dust-sulfate mode explains most of the main development region SST variance at time scales of 20 years or more.

## Discussion

We estimated dust radiative forcing to have contributed −0.06 K to the SST anomaly depression during the 1970s and 1980s, out of a total of −0.13 K for that period, most of which is captured by an SST mode associated with sulfate and dust. By those estimates, dust radiative forcing alone can explain 46% of the average SST decrease during those two decades. Accounting for some uncertainty on the dust reconstruction and the dust optical properties yields bounds of −0.04 and −0.10 K of SST perturbation dust to dust radiative forcing, which can explain 31–77% of the total. The remainder of the SST anomaly may be explained by local sulfate forcing and wind and cloud feedbacks. These results indicate that, in the tropical North-Atlantic, the 1970s–1980s SST depression and the hurricane drought are mostly consistent with those expected from the estimated concurrent dust loading. Based on the idea that, on a larger scale, the dust-sulfate mode and the associated Sahel hydrological drought are likely caused by sulfate forcing (e.g., refs. 36, 62), we propose that a large fraction of the sulfate aerosol emissions on the main development region SST is via their effect on dust lofting and transport. Hence, the dust-sulfate mode is seen here as resulting largely from the dust radiative forcing in the main development region, which acts as positive feedback on regional anthropogenic and volcanic sulfate forcing. This feedback explains why models, most of which do not capture dust variability[28], struggle to capture the tropical North-Atlantic SST depression and the hurricane drought of the 1970s and 1980s.

Individually, the results presented here do not depart much from existing literature. In different sub-fields of climate science, it is fairly well accepted that there was a hurricane drought in the 1970s and 1980s[3], that hurricane activity correlates well with temperature anomalies[1], that Saharan dust variability can explain an important fraction of SST variability[25,63] and hurricane activity variability[27] at long time scales, that dust cover over the tropical North-Atlantic correlates well with dry conditions in the Sahel[43], and that anthropogenic aerosols were an important contributor to hydrological drought in the Sahel in the 1970s and 1980s[38]. However, a mechanism by which anthropogenic sulfate aerosol forcing leads to enhanced dust emissions and a decrease in hurricane activity has, to our knowledge, never been proposed. Such a mechanism, although difficult to verify using model simulations, would help answer outstanding scientific questions regarding hurricane activity past and future, and have important implications that warrant further discussion by the community.

In previous literature, a basin-wide coupled air–sea phenomenon known as the Atlantic Multi-decadal Oscillation (AMO)[64], was also hypothesized to explain multi-decadal variability in tropical North

Atlantic SST anomaly, and the hurricane drought. Our dust-sulfate mode has a very similar structure to this hypothesized oscillation. However, the temporal signature attributed to the AMO in paleoclimate records and CMIP5 preindustrial simulations was recently shown to be due to a multidecadal periodicity in explosive volcanic eruptions[62]. Similarly, 20th-century multidecadal variability attributed to the AMO was shown to be due to anthropogenic[61,65,66] and volcanic[67] radiative forcing. Hence, we did not consider a possible role for natural variability in this paper. The perspective presented here suggests that we should not expect the currently high hurricane activity to decrease in the upcoming decades in association with a return to the negative phase of a natural oscillation. We do not, however, exclude an important role of natural variability at quasi-decadal or shorter time scales. An additional important caveat is that the proposed mechanism does not account for wind and cloud feedbacks, which may also influence the amplitude of the tropical North-Atlantic variability at a long time scales[55].

Tropics-wide warming, since it occurs at the same rate as main development region warming, is unlikely to explain the observed magnitude of power dissipation index variations. However, global warming can cause storms of similar intensity to produce more intense rainfall, which increases their destructiveness, even though it might not affect the power dissipation index[68]. In addition, midlatitude warming could allow category 1 and 2 hurricanes to travel further poleward[69], strongly impacting ill-prepared communities. Since the power dissipation index is dominated by major hurricanes, a poleward shift of weak hurricanes will not be well captured by power dissipation index estimates and is unlikely to depend much on the main development region SST anomaly. In addition, higher frequency natural variability influencing SST must also be considered as influencing power dissipation index, albeit on shorter time scales. Finally, on the centennial timescale, it is likely that the global mode will become important for TC activity.

## Methods

### Power dissipation index

The power dissipation index for a given storm is computed using the IbTRACS dataset[70] and the formula of Emanuel (2007)[1]

$$PDI = \int_0^\tau V_{max}^3 d\tau, \qquad (2)$$

where $V_{max}$ is the maximum velocity within the storm at a given time, and $\tau$ is the duration of the event. Individual storm PDI are then summed over all the storms in a given category or during a given month, to produce Fig. 1. It can be shown[1] that PDI during a given period can be represented as the product between the number of storms during that period $N$, the intensity-weighted duration of the storms $D$, and the averaged intensity of the storms $I$. Hence, $PDI = N \times D \times I$, where

$$D = \frac{1}{N}\sum_{i=1}^{N} D_i, \qquad D_i = \frac{\int_0^\tau V_{max} dt}{V_{lmax}}, \qquad I = \frac{1}{ND}\sum_{i=1}^{N}\int_0^\tau V_{max}^3 d\tau, \qquad (3)$$

and $V_{lmax}$ is the lifetime maximum intensity. When the variations of $I$, $N$ and $D$ between two averaging periods are small enough, we can approximate fractional variations in PDI as the sum of the fractional variations of $I$, $N$ and $D$, so that

$$\frac{\Delta PDI}{PDI} \approx \frac{\Delta N}{N} + \frac{\Delta D}{D} + \frac{\Delta I}{I}. \qquad (4)$$

### Dust proxy

In order to get an estimate of dust optical thickness as a function of time over the main development region during the 20th century, we

combine three different datasets: boundary layer dust concentration measured at Barbados[53], AVHRR satellite measurements of aerosol optical depth over the main development region[22], and a Sahel precipitation index[54] (Joint Institute for the Study of the Atmosphere and Ocean, doi:10.6069/H5MW2F2Q). Timeseries of dust concentration and optical depth are averaged from June to September because, in our simulations, SST takes approximately a month to respond to dust forcing (see the "Methods" subsection "MIT single column model"). The Sahel precipitation index is annually averaged since it is strongly concentrated during the summer when dust concentration is high over the tropical North Atlantic main development region. All timeseries are low-pass filtered with a 7 years period.

First, we estimate dust optical depth over the main development region from 1965 to 2012. To do so, because the AVHRR record begins in 1982 and because dust concentration peaks in the early 1980s, we rescale the Barbados dust concentration timeseries so that dust optical depth in the 5 years period from 1982 to 1987 in the rescaled timeseries is the same as in the AVHRR dust optical depth peak averaged over the main development region for that period. This method, which assumes that zero dust concentration at Barbados implies zero dust optical depth over the main development region, is similar to the method used by Strong et al.[27,63]. A caveat is that the AVHRR data exhibits substantial uncertainties and appears to underestimate optical depths in the main development region by comparison to the more recent MODIS satellite products[56]. We cannot use MODIS data for this rescaling because it does not extends sufficiently far back in time, but only to 2001. Hence, this rescaling yields a coarse estimate of how dust optical depth may have varied over the main development region since the 1960s, a time period that is not covered by the satellite measurements. Then, similarly to Mahowald et al. (2010)[47], to further extend the dust record in time, we compute a regression of Barbados dust on a Sahel precipitation index (mm/month) extending from 1901 to 2017. Using this method, we estimate that the average dust optical depth over the main development region in the 1970s and 1980s was about 0.230 and that the average dust cover over the 1960–2017 period where we can reasonably trust hurricane observations was about 0.187 so that dust optical depth during the hurricane drought was about 0.043 higher on average. To assess the uncertainty of this estimate, we compute a bias-corrected and accelerated (BCa) bootstrap confidence interval on the regression coefficient between Sahel precipitation index and the Barbados-based dust optical depth record using 1000 data samples and obtain an interval of $0.043 \pm 0.010$. Using the Wald method independently yields a very similar estimate of the confidence interval. Since we do not explicitly compute an estimate of the uncertainty on the Barbados-based optical depth record, this uncertainty is likely an underestimate. We also note that these estimates are sensitive to the definition of the main development region.

### Sulfate asymmetry index
The aerosol optical depth asymmetry is an ad hoc index for the radiative forcing, computed from GISS-E2-1-G climate model[71,72] tropospheric sulfate aerosols along with the corresponding stratospheric aerosol data set[73,74]. The index is computed by taking the difference in average sulfate optical depth between a northern and a southern region, both extending from 35W and 55E, to focus on Africa and Europe. The southern region extends from 60S to the equator and the northern region from the equator to 60N. The index is not very sensitive to those bounds. Since aerosol loading is generally low in the southern region, the value of the index is similar to the average optical depth in the northern region.

### Low-frequency component analysis
Since forced SST response in the Atlantic is thought to have longer time scales than natural variability[17], we aim to isolate low-frequency variability modes by using the low-frequency component analysis method (described in detail in ref. 59). This method finds linear combinations of the leading empirical orthogonal functions in a dataset, that maximize the ratio of low-frequency to total variance while retaining variance at all frequencies. This method can be thought of as a signal-to-noise maximizing empirical orthogonal functions analysis method (e.g., ref. 75). The results are low-frequency patterns (LFPs), which are not orthogonal to one another, and the associated low-frequency components (LFCs), which are uncorrelated. The resulting modes are sorted by the ratio of low-frequency to the total variance, so we expect the first LFC, named the global mode here, to have the longest time scale. This method has two input parameters: the low-pass filter cutoff, used to define low-frequency, and the number of leading empirical orthogonal functions used. Here, we set the low-pass filter cutoff at 7 years, to filter out interannual variability in the Atlantic, and the number of leading empirical orthogonal functions to 25, based on previous literature[59]. The results are not very sensitive to the choice of parameters. This method has been shown to capture accurately both forced (global warming) and natural (El-Nino Southern Oscillation) modes of variability, and we apply it here to see whether specific modes of variability of the tropical North Atlantic SST can be related to the sulfate aerosols and the dust feedback. We analyze SST data from the Hadley Centre Global Sea Ice and Sea Surface Temperature (HadISST) dataset[76] from 1870 to the present. The mean seasonal cycle is removed from the data, but the global warming signal is not.

### MIT single-column model
In order to see if the amplitude of the main development region SST anomaly can be attributed to dust, we run simulations using the MIT SCM[77] under a weak temperature gradient (WTG) constraint, with added aerosols. Under the WTG approximation, temperatures in the free troposphere above the boundary layer are held constant, assuming both that the region of perturbed SST covers a small fraction of the tropics and that atmospheric dynamics homogenizes temperature on pressure surfaces on short time scales in the tropics[78]. The boundary layer height above which WTG is applied is chosen similarly to the previous literature[79] to be 850 hPa, which is reasonable over the tropical oceans. The simulated sensitivity of SST to optical thickness can be almost an order-of-magnitude larger when running simulations in radiative-convective equilibrium, but the WTG constraint is more appropriate to simulate the effects of dust over a limited area of ocean[10,80,81]. We use the MIT SCM along with the two-spectral-intervals radiation scheme of Fouqart[82] and Morcrette[83]. The model uses the convection parameterization of Emanuel and Zivkovic-Rothman[84] to compute the evolution of water vapor. The simulations do not allow for cloud feedback on radiation and instead use fixed profiles of cloud fraction to decrease the noisiness of the sensitivity experiments. Despite this simplification, we acknowledge that the feedback of clouds in WTG is an important topic in its own right which may be relevant to main development region SSTs, as suggested by observations[55]. In the simulations, the ocean is represented by a slab, the temperature of which varies in response to changes in the balance between net radiation and turbulent enthalpy fluxes at the surface. Possible effects of ocean heat flux convergence are set to zero. We choose a 25 m slab thickness, which is representative of summer mixed layers depths[25,85]. This choice has no influence on equilibrium perturbation SST, but only on the time, it takes to reach this equilibrium. In our WTG-constrained simulations, the SST perturbation reaches 75% of its equilibrium value after a month, and over 95% of its equilibrium value after 2 months, much faster than radiative-convective equilibrium simulations. We note that while our simulations parameterize the effects of gravity waves radiating local temperature perturbations away—an important constraint by the large-scale environment on the response to a local forcing—they do not represent other potentially important environmental responses to a local forcing such as possible changes in near-surface winds.

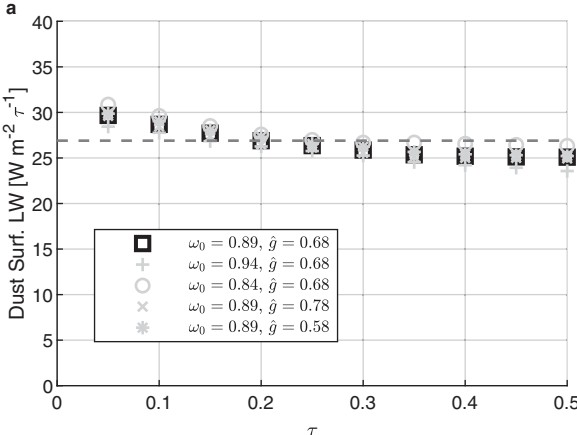

**Fig. 5 | Dust radiative forcing efficiency in the single column model and in observations.** Panel **a** Simulated dust longwave forcing efficiency at the surface (black squares), and estimate from Song et al.[86] (dashed black line). Panel **b** Simulated dust shortwave forcing efficiency at the surface (black squares), and estimate from Song et al.[86] (dashed black line), and at the top-of-atmosphere (TOA, blue squares), and estimate from Song et al.[86] (dashed blue line). The faded profiles represent the simulations with $\hat{g}$ between 0.58 and 0.78 and $\omega_0$ between 0.84 and 0.94.

To gain insight into the sensitivity of SST, potential intensity, and other TC-relevant quantities like the genesis potential index[57] to Saharan dust optical thickness, we adjust the simulated dust shortwave (SW) asymmetry parameter ($\hat{g}$) and single scattering albedo ($\omega_0$), and the longwave (LW) extinction efficiency of dust to produce similar surface and top-of-atmosphere (TOA) forcing efficiencies as observed by Song et al.[86]. Forcing efficiency is defined as the aerosol direct radiative forcing, normalized by the optical thickness ($\tau$) at 0.55 μm (e.g., ref. [87]). We take $\hat{g}$ = 0.68 and a single scattering albedo of $\omega_0$ = 0.89 for SW. We retain the assumption of the scheme of Morcrette[83] that aerosols do not scatter LW and set $\tau_{10\mu m}/\tau_{0.55\mu m}$ = 0.45. These parameters are broadly consistent with dust observations and, more importantly, yield similar dust forcing efficiencies as those derived from observations. As a reference and to test the sensitivity of the results to the dust optical properties, we vary $\hat{g}$ between 0.58 and 0.78 and $\omega_0$ between 0.84 and 0.94, both unlikely low and unlikely high values for dust the asymmetry parameter and the single-scattering albedo. The vertical dust profile is Gaussian shaped with a maximum at 700 hPa and a standard deviation (controlling the width of the Gaussian function) of 100 hPa. This is an idealized version of the Saharan air layer dust profile as it moves from West Africa over the ocean (see e.g., ref. [88]). As the dust moves further West the altitude of the layer decreases progressively a bit, which might impact its longwave radiative effects.

Figure 5 shows the forcing efficiency of the prescribed aerosols as a function of optical thickness, which is varied from 0 to 0.5 in increments of 0.05. Black squares denote a surface forcing efficiency and blue squares a TOA efficiency. The horizontal dashed lines are state-of-the-art estimates of dust forcing efficiency computed by Song et al.[86], based on particle size distributions from the 2011 Fennec aircraft campaign[89,90], refractive indices from Colarco et al.[91] in the SW and Di Biagio et al.[92] in the LW, and particle shape distributions from Dubovik et al.[93]. At dust concentrations similar to those over the MDR ($\tau \approx 0.2$), the forcing efficiencies produced by the MIT SCM are within 10% of the state-of-the-art estimates of Song et al.[86]. Considering different models for particle size distribution, refractive indices and particle shape distributions, yields a broader range of possible radiative forcing efficiencies than the ones simulated here[86]. This good correspondence with observation, which is a consequence of purposely adjusting $\hat{g}$, $\omega_0$ and $\tau_{10\mu m}/\tau_{0.5\mu m}$ suggests that the simulated sensitivity of SST to dust optical thickness should be close to real-world values.

## Parseval's theorem

Parseval's theorem is used to compute the fraction of the SST variance that is associated with periods between 20 and 100 years. 20 years was chosen as a lower period bound because it is the duration of the hurricane drought. In effect, we compute a multidecadal-to-total variance ratio given by

$$F_v = \frac{\int_{f_l}^{f_h} \text{PSD}\, df}{\int_{f_0}^{f_{nyq}} \text{PSD}\, df}, \tag{5}$$

where $f_l = 1/100$ year$^{-1}$, $f_h = 1/20$ year$^{-1}$, $f_0$ is the lowest frequency that can be captured by the timeseries, $f_{nyq}$ is the Nyquist frequency, and PSD is a power spectral density estimate computed using Thomson's multitaper method[94]. First, the theorem is applied to the main development region SST anomalies, and then to the main development region SST anomalies minus the contribution of the dust-sulfate mode. The relative difference indicates that the dust-sulfate mode contributes 88% of the SST anomaly variance at multidecadal time scales.

## Data availability

The dataset generated using the MIT single column model has been deposited in the Zenodo database under accession code doi:10.5281/zenodo.6423326. The HadISST dataset[76] (doi: 10.5065/XMYE-AN84) is available from the MET Office. The SSPI dataset[54] (https://doi.org/10.6069/H5MW2F2Q) is available from the Joint Institute for the Study of the Atmosphere and Ocean. The GISS simulation dataset[71,72] (https://doi.org/10.22033/ESGF/CMIP6.7127) of tropospheric sulfate aerosols is hosted on https://esgf-node.llnl.gov/. The dataset of stratospheric volcanic aerosols prescribed in the GISS simulations[73,74] (https://doi.org/10.5067/GLOSSAC-L3-V1.0) is hosted on ftp://iacftp.ethz.ch/pub_read/luo/CMIP6/. The Barbados dust dataset[43] is curated by Joseph Prospero and is available upon request. The AVHRR dust optical depth dataset[22] is curated by Amato Evan and is available upon request.

## Code availability

The MIT single column model source code and the scripts used in processing the data and in producing all figures have been deposited in the Zenodo database under accession code doi:10.5281/zenodo.6423326. The analysis scripts are sorted in folders named after the figures produced by each script.

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

## Acknowledgements

The work presented here is based in part on the Ph.D. thesis of Raphaël Rousseau-Rizzi[7]. This work was supported by the NSF grant AGS-1906768 attributed to K.E. Support for the Twentieth Century Reanalysis Project version 3 dataset is provided by the U.S. Department of Energy, Office of Science Biological and Environmental Research (BER), by the National Oceanic and Atmospheric Administration Climate Program Office, and by the NOAA Physical Sciences Laboratory. The authors would also like to thank Megan Lickley, Tom Beucler, and Amato Evan for their advice.

## Author contributions

R.R.R. and K.E. designed the study, R.R.R. performed the analyses. R.R.R. and K.E. contributed to the interpretation and discussion of the results.

## Competing interests

The authors declare no competing interests.

## Additional information

**Supplementary information** The online version contains

supplementary material available at https://doi.org/10.1038/s41467-022-32779-y.

