## [Peer Review File · Nature Communications]

Natural and Anthropogenic Contributions to the Hurricane Drought of the 1970s-1980sREVIEWER COMMENTS

Reviewer #1 (Remarks to the Author):

Review's comments for the paper (NCOMMS-21-35054-T), entitled "Natural and Anthropogenic Contributions to the Hurricane Drought of the 1970s-1980s", submitted to Nature Communications

Understanding the decadal-multidecadal variability of Atlantic hurricane activity is a hot topic in the literature. By using both observations and model simulations, this paper proposed a mechanism for the decreased Atlantic hurricane activity during the 1970s and 1980s. The research further demonstrated that the radiative effects of sulfate aerosols from Europe and North America decreased precipitation in the Sahara-Sahel region, leading to an enhancement of dust regional emissions and transport over the Atlantic. This dust in turn enhanced the local decrease of sea-surface temperature and of hurricane activity. Results are interesting and the paper is worth of publishing. However, the paper needs some improvements by properly addressing the comments listed below before it can be accepted for publication in Nature Communications.

Major comments

1. One major concern the reviewer has is that there is already quite extensive research in this topic in the literature. What is the originality and novelty of this paper? It seems to reviewer that authors did not do a good job on the literature review and some important studies on decadal-multidecadal variability of Atlantic hurricane activity based on observations, such as Nyberg et al. (2007) and Vecchi et al. (2021), were missed in the current version. This led to the impression that the decreased Atlantic hurricane activity in 1970s and 1980s was identified in this study.

Nyberg, J., Malmgren, B., Winter, A. et al. Low Atlantic hurricane activity in the 1970s and 1980s compared to the past 270 years. Nature 447, 698–701 (2007).

<https://doi.org/10.1038/nature05895>

Vecchi, G.A., Landsea, C., Zhang, W. et al. Changes in Atlantic major hurricane frequency since the late-19th century. Nat Commun 12, 4054 (2021).

<https://doi.org/10.1038/s41467-021-24268-5>

2. As shown in Introduction, there are already some studies about the Sahara-Sahel dust on reduced Atlantic hurricane activity in 1970s and 1980s. What are new aspects or new advances of the current study in comparison with those cited studies? How the main conclusions about the role of dust in tropical Atlantic SSTs compare with other studies? This shall be better illustrated and demonstrated.

3. Some of key statements on changes between two periods are better to be quantified.

Minor comments

1. Line 4 in paragraph 2 on page 1. "TC" is not defined yet.

2. Lines 3-6 in paragraph 1 on page 3. These statements are about results from CMIP5 type climate models. How are about CMIP6 models? Is there any study using CMIP6 models?

3. Lines 9-20 in paragraph 1 on page 3. Are these statements about previous studies. If they are, please give references.

4. Lines 5-8 in paragraph 2 on page 3. Are CMIP6 better in simulating multidecadal SST variability on the tropical Atlantic?

5. Line 2-3 in paragraph 3 on page 4. "Sahel precipitation index" is shown in Figure 2a. However, its definition is given in lines 1-2 in paragraph 4 on page 4.

6. Lines 6-9 in in paragraph 2 on page 4. Give references for this statement.

7. Figure 2 caption. It seems to the reviewer that blue lines are the same in panels a and b. However, it lacks unit and magnitude information in panel a.

8. Line 4 from the bottom on page 5. "0.1 K depression over those 20 years". How does

this magnitude of cooling compare with observed cooling?

9. Lines 11-14 in paragraph 4 on page 7. "Since the SST modes are uncorrelated to one another, this result suggests a relation between the second mode specifically and Sahel precipitation, or a common driver, and suggests that this mode is associated with dust emissions and radiative forcing". The reviewer cannot understand how authors got these conclusions from Figure 4.

10. 4.1 on page 10. The definition of "Sulfate asymmetry index" is not clear. Is it the area mean over the whole region, or the difference of area mean between the northern part and southern part over the region? Please clarify.

11. Section "MIT single column model". Lines 4-5, "WTG approximation, tropospheric temperatures above 850 hPa are held constant". Why do authors choose 850 hPa? This level might be affected by land-sea contrast. How results are sensitive if you chose 500 hPa?

12. Lines 10-11 on page 12. "a standard deviation of 100 hPa". Meaning is not clear. Please clarify.

Typos

1. Line 2 in paragraph 2 on page 7. "(Fig. 4b))" to "(Fig. 4b)".

2. Line 6 in paragraph 2 on page 7. "(Fig. 4a))" to "(Fig. 4a)".

3. Line 1 in paragraph 4 on page 7. "Figure. 4c)" to "Figure 4c".

4. Line 10 in paragraph 4 on page 7. "(Fig. 4d))" to "(Fig. 4d)".

Reviewer #2 (Remarks to the Author):

Review of "Natural and anthropogenic contributions to the hurricane drought of the 1970s-1980s" by Rousseau-Rizzi and Emanuel

This study looks at the contributions of African dust aerosols and sulfate aerosols to tropical Atlantic SST, and hence hurricane activity, since the 1970s. The topic in general has received a lot of attention in the past 10-15 years. The main idea of the current study is the proposed dominance of dust forcing on tropical North Atlantic SST, combined with the hypothesized connection between sulfate and African dust aerosol contributions, through sulfate-forced changes in Sahel rainfall/winds. However, these links are not demonstrated convincingly. The result is a manuscript in which all of the main pieces (importance of sulfate and dust aerosol forcing on Atlantic SST, importance of multidecadal variations of SST for hurricane activity) have already been shown in previous papers. Significant additional work would be needed to test the proposed ideas and come up with novel and robust results.

Specific comments:

GPI discussion: Observed TC activity was 80% lower in 1970-90 compared to 1990-2010, yet even a very large increase in dust optical depth of 0.3 would result in only a 55% change in TC genesis. So the observed change in DAOD (and resultant change in SST) contributed only about 10-15% of the TC genesis reduction. Other factors like shear and vorticity must be important, and its unclear if they are related to dust-induced changes in SST.

More details are needed about how the surface radiation from the single column model forces the ocean. Is there any coupling between the atmosphere and ocean (damping of SST through surface turbulent heat fluxes)? Or is this done through damping to climatological SST (or something else)? A 2-m deep slab mixed layer will be extremely sensitive to changes in the surface radiative heat flux. How does your dust-forced SST anomaly vary with changes in mixed layer depth? Fig. 3: Why are SSTs 30-31C in (a)? That's unrealistically high.

It's hard to believe that increased dust radiative forcing, which it is claimed was

ultimately driven by higher levels of sulfate aerosols (through their impact on SST -> atmospheric circulation -> Sahel precipitation/winds), could be the main driver of lower SST during the 1970s and 80s. You're saying that a 0.03C decrease in SST due to sulfate aerosols caused a 0.1C decrease in SST due to dust, if I understand correctly. That would mean that Sahel rain/winds/dust are extremely sensitive to tropical Atlantic SST, which has not been shown. That is the weakest part of the argument. Since you are claiming that almost all of the negative SST anomaly during the 70s and 80s was caused by sulfate and dust aerosols, and the positive dust anomalies were driven by sulfate forcing, the impact of sulfate aerosols (and/or tropical Atlantic SST) on dust optical depth needs to be quantified convincingly. Based on the current arguments in the manuscript, it seems more likely that the dust radiative forcing has been overestimated, or natural variability in the ocean (AMOC, NAO, etc.) was an important driver of the cooler SSTs during the 70s and 80s (or there is a natural multidecadal cycle of dust emission (unrelated to SST forcing) that dominates the radiative forcing of tropical North Atlantic SST).

It's already been shown that African dust has contributed to decadal-multidecadal variations of tropical Atlantic SST and hurricane activity (Evan et al., *Nat. Geosci.*, 2011; Wang et al., *J. Clim.*, 2012), so that is not new. It's also known that sulfate aerosols have had a big impact on tropical Atlantic climate, and it's been argued that we are currently in a negative phase of the AMO due to anthropogenic aerosol and GHG forcing (Mann et al., *GRL*, 2014). The new idea in the present study seems to be the connection/interaction between sulfate and dust aerosols. However, more robust connections between aerosols, SST, and hurricane activity need to be established in order to make the present study novel and convincing enough to be published.

Reviewer #3 (Remarks to the Author):

Review of "Natural and Anthropogenic Contributions to the Hurricane Drought of the 1970s-1980s" by Raphael Rousseau-Rizzi and Kerry Emanuel, for Nature Communications

This paper explores the role of dust feedbacks in driving the amplitude of MDR SSTs, and hence Atlantic hurricane activity, during the 1970s-1980s. The authors suggest that anthropogenic aerosol emissions drove a reduction in Sahel precipitation, which then led to a decrease in dust transport across the tropical Atlantic and that it is this feedback that was responsible for the majority (77%) of the observed MDR SST cooling during this period. This is an interesting paper, and an important topic, but I think the authors need to provide a clearer narrative about the novelty of their results, a clearer explanation of methods used and finally propagate the uncertainties in their analysis. Hence I recommend major revisions in the hope that these issues can be addressed.

I think it is well known that decreases in Sahel precip during the 1970s-1980s occurred coincident with an increase in dust emissions from the region and that this increases the dust optical depth across the MDR region of the tropical North Atlantic and that this *potentially* offers a dust feedback onto MDR SSTs and hence North Atlantic tropical cyclone activity. In particular, I am aware that Amato Evan has written a number of papers on the topic over the years (which you cite). So I think the particular novelty of this paper needs to be clearly framed in this context.

This paper attempts to quantify the strength of the dust feedback and hence examine the extent to which dust feedbacks are necessary in order to realise the amplitude of MDR SST cooling during the 1970s-1980s. However, there are many other mechanisms that could explain the MDR SST, for example the circulation changes linked to reduced Sahel rainfall (i.e. the African monsoon/ITCZ) may also change the low-level winds over the MDR that then modify the SSTs via a wind-evaporation-cloud mechanism – this mechanism might also not be well captured by present climate models. Furthermore, the direct aerosol forcing mechanism might itself not be well captured by current climate

models – there is large uncertainty in regional aerosol forcing and the simulation of aerosol-cloud indirect effects. Finally, there is quite a lot of evidence to suggest that models may underrepresent the strength of the connection between the North Atlantic sub-polar gyre and tropical SSTs – the AMV horseshoe does not extend far enough south (or weakens too much). Whilst I am not saying that the authors need to go into any detail on these alternative hypotheses, I think it does make it essential to outline the uncertainties in their analysis so readers can make their own assessment on the confidence in these results – something that I feel is missing at the moment.

The key result, upon which the paper conclusions are based, is that the dust optical depth change in the 1970s-1980s is estimated to be ~ 0.07 which then is then calculated to lead to a reduction in MDR SST of 0.1 K. But I do not think the method to arrive at this is fully explained and the uncertainties are not propagated. It is not clear to me how the Barbados optical depth timeseries in Fig. 2a is arrived at. The Methods section also seems to lack detail – I think the satellite measurements need to be shown in some way to see how they are used to ‘rescale’ the Barbados measurements and hence arrive at the 0.07 optical depth change and crucially what the estimated confidence interval is on this estimate. Then the single-column model simulations could also be better explained, including key assumptions and limitations. There is a brief discussion of the impact of modifying the dust optical properties, but this sensitivity is not followed through to an uncertainty in the 0.1K, or the final 77% figure. So essentially, there should be confidence intervals on the optical depth, the sensitivity of SST to SW, and hence the SST reduction and fraction of variance explained. I think if these are propagated through then there would be quite a large uncertainty associated with the impact of dust feedbacks. I am not saying that this in anyway invalidates the results presented, but it might have implications for the conclusions drawn by the reader when comparing it to other evidence.

I also wonder if there other lines of evidence that the authors could use to add weight to their analysis. For example, does natural interannual variability in Sahel precip driven by ENSO give the opportunity to validate some of the dust optical properties and SST sensitivity using more modern satellite observations, or ground data? Or, does the spatial pattern of dust induced SST variability provide a key fingerprint?

Part of the motivation appears to be that present climate models do not simulate the amplitude of MDR SST cooling in the 1970s-1980s, but the authors do not provide direct evidence of this, or even quantify this from the cited literature and furthermore I’m not sure the references cited show this clearly. The authors suggest that models ‘account for sulfate radiative forcing’ but we do not know the extent to which they model the processes correctly.

I don’t think the role of anthropogenic aerosols in driving Sahelian dust variability is well described. As I see it there are two possible mechanisms: i) that European and North American anthropogenic aerosol emissions (mainly sulfate) drove interhemispheric temperature gradients that drove meridional shifts in the ITCZ, including over the Sahel, leading to the reduce Sahel precip. ii) that European aerosols alone are advected south, reduce surface SW heating over North Africa directly weakening the Saharan heat low and the West African Monsoon resulting in a decrease in Sahel precipitation. From the paper I’m not sure which of these mechanisms the authors are appealing to? See Dong et al 2014 for the latter in an atmosphere only climate model experiment (<https://journals.ametsoc.org/view/journals/clim/27/18/jcli-d-13-00769.1.xml>)

Response to Reviewers: “Natural and Anthropogenic Contributions to the Hurricane Drought of the 1970s-1980s”

Raphaël Rousseau-Rizzi and Kerry Emanuel

General response

We thank the reviewers for their very insightful and valuable comments, and for all of their work in reviewing and helping to improve this paper. Following the reviewers’ comments, instead of comparing the hurricane drought of the 1970s and 1980s to the following two decades of recovery, we are now comparing the hurricane drought to the longer period of 1960 to 2017 which starts when hurricanes became better-observed. Because of this change, most results reported in the paper are numerically different than in the previous submission, but their interpretation remains similar.

In our opinion, the aspect that stands out the most, from all three reviews, is the question of the novelty of the work presented in this study. Each review focuses more specifically on a different research topic, that the conclusions of our manuscript depend upon. More precisely, reviewer 1 points out that the hurricane drought of the 1970s and 1980s has already been reported and discussed in the literature (although its causes are still unclear), reviewer 2 states that regional climate influences of sulfates and dust on SST, as well as the connection between hurricane activity and SST at long time scales have already been studied extensively, and some concerns of reviewer 3 revolve around the fact that the relation between dust emissions and drought is well documented.

We agree with all the reviewers that no single analysis the paper we submitted substantially pushes the limits of our knowledge in the research topics mentioned above. Correspondingly, we do not mean to argue that the value of this paper lies with any individual novel result - although some analyses, like our use in this context of the single column model, are novel - and we make sure to clarify this in the revised version of the paper.

However, we do propose a completely new explanation for the hurricane drought, that arises from integrating multiple sub-fields within climate science, and we think that this is very valuable. This explanation has, to our knowledge, never been mentioned in a publication, and one of the key aims of our paper is to get the community to discuss and evaluate it. The very reviews we received confirm the value of this aim since the reviewers seem to have different research specialties and focus on different aspects of the paper. From the perspective of a scientist studying hurricane activity, this paper is meant to propose an explanation for the hurricane drought that includes an explanation for observed dust variations. From

the perspective of a someone studying dust variability, this paper aims at highlighting the cascading impacts an anthropogenic influence on dust variations may have.

Reviewer 1

Understanding the decadal-multidecadal variability of Atlantic hurricane activity is a hot topic in the literature. By using both observations and model simulations, this paper proposed a mechanism for the decreased Atlantic hurricane activity during the 1970s and 1980s. The research further demonstrated that the radiative effects of sulfate aerosols from Europe and North America decreased precipitation in the Sahara-Sahel region, leading to an enhancement of dust regional emissions and transport over the Atlantic. This dust in turn enhanced the local decrease of sea-surface temperature and of hurricane activity. Results are interesting and the paper is worth of publishing. However, the paper needs some improvements by properly addressing the comments listed below before it can be accepted for publication in Nature Communications.

Major comment 1

One major concern the reviewer has is that there is already quite extensive research in this topic in the literature. What is the originality and novelty of this paper? It seems to reviewer that authors did not do a good job on the literature review and some important studies on decadal-multidecadal variability of Atlantic hurricane activity based on observations, such as Nyberg et al. (2007) and Vecchi et al. (2021), were missed in the current version. This led to the impression that the decreased Atlantic hurricane activity in 1970s and 1980s was identified in this study

Nyberg, J., Malmgren, B., Winter, A. et al. Low Atlantic hurricane activity in the 1970s and 1980s compared to the past 270 years. *Nature* 447, 698–701 (2007). <https://doi.org/10.1038/nature05895>
Vecchi, G.A., Landsea, C., Zhang, W. et al. Changes in Atlantic major hurricane frequency since the late-19th century. *Nat Commun* 12, 4054 (2021). <https://doi.org/10.1038/s41467-021-24268-5>

We completely agree that there is exhaustive literature on the topic, and that the hurricane drought of the 1970s-1980s has already been documented, among others by Nyberg et al. 2007. It is not our intention to propose that we are the first to report the hurricane drought and we agree with the reviewer that this should be made clearer. To remedy that, we have added, at the end of the very first sentence of the paper, references to three previous papers that either report or discuss the hurricane drought. We included the two references suggested by the reviewer as well as citing Emanuel (2021), which offers counterarguments to Vecchi

et al. (2021).

Emanuel, K., 2021. Atlantic tropical cyclones downscaled from climate reanalyses show increasing activity over past 150 years. Nature communications, 12(1), pp.1-8.

Because of the limits in Nature Communications on the number of citations (about 50 for the main body of the text) the number of citations for this paper had been pared down from about 120 to about 80, including methods. Given the large number of topics referenced in this paper, this may explain the sparsity of citations on each specific topic relating to the hurricane drought.

We also do agree with the reviewer that most analyses included in this work simply verify and combine individual results presented in the literature. We expand upon the novelty of the research and comparisons to previous literature below in the response to major comment no.2

Major comment 2

As shown in Introduction, there are already some studies about the Sahara-Sahel dust on reduced Atlantic hurricane activity in 1970s and 1980s. What are new aspects or new advances of the current study in comparison with those cited studies? How the main conclusions about the role of dust in tropical Atlantic SSTs compare with other studies? This shall be better illustrated and demonstrated.

We agree that this needs to be better clarified. The fact is that most of our results do not depart much from existing literature. The novelty of this paper arises from comparing and contrasting different sources of evidence already present in the literature, which we support with our own analyses (see *General response* section at the beginning of the document). We clarify this point in a the following paragraph, which we added to the paper in the discussion section:

“Individually, the results presented here do not depart much from existing literature. In different sub-fields of climate science, it is fairly well accepted that there was a hurricane drought in the 1970s and 1980s (Nyberg et al., 2007), that hurricane activity correlates well with temperature anomalies (Emanuel, 2007), that Saharan dust variability can explain an important fraction of SST variability (Evan et al., 2012; Strong et al., 2015) and hurricane activity variability (Strong et al., 2018) at long time scales, that dust cover over the tropical North-Atlantic correlate well with dry conditions in the Sahel (Prospero and Lamb, 2003), and that anthropogenic aerosols were an important contributor to drought in the Sahel in the 1970s and 1980s (Ackerley et al., 2011). However, a mechanism by which anthropogenic

sulfate aerosol forcing leads to enhanced dust emissions and a decrease in hurricane activity has, to our knowledge, never been proposed. Such a mechanism, although difficult to verify using model simulations, would help answer outstanding scientific questions regarding hurricane activity past and future, and have important implications that warrant further discussion by the community.”

Major comment 3

Some of key statements on changes between two periods are better to be quantified.

We agree that the periods were not clearly defined. The two periods we are comparing have now been redefined to be the 1970s and 1980s and the period extending from 1960 to 2017. Changes between the periods have also been further elaborated upon, including a breakdown of the components of PDI change between the two periods into storm number, storm duration and storm intensity.

Minor comment 1

Line 4 in paragraph 2 on page 1. “TC” is not defined yet.

Agreed. We have now defined it.

Minor comment 2

Lines 3-6 in paragraph 1 on page 3. These statements are about results from CMIP5 type climate models. How are about CMIP6 models? Is there any study using CMIP6 models?

We agree that some information regarding state-of-the-art CMIP6 simulations should be provided. Yes, there are CMIP6 individual-model studies showing the underestimation of multidecadal SST variability in the main development region for MIROC (Watanabe and Tatebe, 2019), NorESM (Vågane, 2020) and CNRM (Voltaire et al., 2019). There are also studies showing the underestimation of multidecadal SST variability at the basin scale for HadGEM (Andrews et al., 2020), UKESM (Robson et al., 2020) and IPSL (Boucher et al., 2020). We have not found studies comparing multiple CMIP6 models, perhaps because they have simply not been published yet. To clarify this point, in the revised paper, we cite the studies that show an underestimate in the tropical North-Atlantic and the main development region. The statement are now:

“However, despite accounting for sulfate radiative forcing, climate model simulations fail

to represent the magnitude of SST variations at multidecadal time scales, in the tropical North-Atlantic and the main development region (Bellomo et al., 2018; Voldoire et al., 2019; Watanabe and Tatebe, 2019; Vågane, 2020). As a result, models also fail to capture the full observed depression of hurricane activity of the 1970s and 1980s (Dunstone et al., 2013; Villarini and Vecchi, 2013).”

Minor comment 3

Lines 9-20 in paragraph 1 on page 3. Are these statements about previous studies. If they are, please give references.

We agree that this was unclear. These statements were not about previous studies and we have clarified this by rephrasing: “We propose that Saharan dust emissions could provide such a feedback.”

Minor comment 4

Lines 5-8 in paragraph 2 on page 3. Are CMIP6 better in simulating multidecadal SST variability on the tropical Atlantic?

The CMIP6 studies we found tend to focus on single models (see response to Minor comment 2) and still fall short of simulating the observed variability. However, we don't have a good sense whether they are collectively better or worse than CMIP5 at simulating multidecadal SST variability.

Minor comment 5

Line 2-3 in paragraph 3 on page 4. “Sahel precipitation index” is shown in Figure 2a. However, its definition is given in lines 1-2 in paragraph 4 on page 4.

Agreed. We moved the definition of the Sahel precipitation index averaging region to the previous paragraph.

Minor comment 6

Lines 6-9 in in paragraph 2 on page 4. Give references for this statement.

This statement is not based on the literature but on the analysis presented in the paper. We

agree that this was unclear and it has been clarified by adding “In our analysis [...]” at the beginning of the statement.

Minor comment 7

Figure 2 caption. It seems to the reviewer that blue lines are the same in panels a and b. However, it lacks unit and magnitude information in panel a.

The two blue lines are indeed the same, except that the one in 2a has been rescaled to become the dust optical depth reconstruction. In 2a, it represents dust optical depth, and its magnitude is indicated on the left axis. To clarify this, we have added: “The precipitation timeseries is the same as in Fig. 2a, except it is not rescaled here.” to the description of panel 2b.

Minor comment 8

Line 4 from the bottom on page 5. “0.1 K depression over those 20 years”. How does this magnitude of cooling compare with observed cooling?

Since we changed the reference period used in this paper, the results are numerically different, but we agree that a comparison to observed cooling needed to be added and we did (see paragraph 1 of the discussion).

Minor comment 9

Lines 11-14 in paragraph 4 on page 7. “Since the SST modes are uncorrelated to one another, this result suggests a relation between the second mode specifically and Sahel precipitation, or a common driver, and suggests that this mode is associated with dust emissions and radiative forcing”. The reviewer cannot understand how authors got these conclusions from Figure 4.

We agree that this is unclear. This statement was meant to be supported by the previous sentence, not Figure 4 per se, and we tried to rephrase it and make this clearer. These sentences have been changed to:

“Further, the second low-frequency component (Fig. 4d) is the only component that correlates well with Sahel drought ($R = 0.69$). The fact that no other SST mode correlates well with Sahel drought suggests a relation between the second mode specifically and Sahel precipitation, or a common driver, and suggests that this mode is associated with dust

emissions and radiative forcing.”

Minor comment 10

4.1 on page 10. The definition of “Sulfate asymmetry index” is not clear. Is it the area mean over the whole region, or the difference of area mean between the northern part and southern part over the region? Please clarify.

We agree and we have clarified the definition, which is now: “The index is computed by taking the difference in average sulfate optical depth between a northern and a southern region, both extending from 35 W and 55 E, to focus on Africa and Europe. The southern region extends from 60 S to the equator and the northern region from the equator to 60 N. The index is not very sensitive to those bounds. Since aerosol loading is generally low in the southern region, the value of the index is similar to the average optical depth in the northern region.”

Minor comment 11

Section “MIT single column model”. Lines 4-5, “WTG approximation, tropospheric temperatures above 850 hPa are held constant”. Why do authors choose 850 hPa? This level might be affected by land-sea contrast. How results are sensitive if you chose 500 hPa?

In the WTG approximation, it is assumed that temperature gradients are small in the free troposphere. Typically, 850 hPa is chosen as the top of the boundary layer, and the height above which the WTG approximation is applied (Shaevitz and Sobel, 2004). We agree that, in certain situations where the air is extremely dry, the top of the boundary layer may be as high as 500 hPa (like over the Saharan desert), but over the ocean, that is much larger than observed boundary layer heights. In addition, applying WTG above 850 hPa constrains the atmosphere more strongly than applying it above 500 hPa only, and hence makes the SST less sensitive to the forcing (E.g., the discussion in Emanuel and Sobel, 2013). In other words, the SST sensitivity to dust is smaller because of the WTG constraint, and either relaxing that constraint or applying it higher up in the atmosphere would just make our results apparently stronger.

We do agree that the choice of 850 hPa needs to be clarified, and the text describing the use of the WTG approximation has been modified to “Under the WTG approximation, temperatures in the free troposphere above the boundary layer are held constant, assuming both that the region of perturbed SST covers a small fraction of the tropics and that atmospheric dynamics homogenizes temperature on pressure surfaces on short time scales in the tropics (Sobel and Bretherton, 2000). The boundary layer height above which WTG is applied is

chosen similarly to previous literature (Shaevitz and Sobel, 2004) to be 850 hPa, which is reasonable over the tropical oceans.”

Minor comment 12

Lines 10-11 on page 12. “a standard deviation of 100 hPa”. Meaning is not clear. Please clarify.

We agree that this was unclear. We were referring to the width of the Gaussian shaped dust profile, and the parameter controlling that width is the standard deviation. We have clarified this and the sentence now reads: “The vertical dust profile is Gaussian shaped with a maximum at 700 hPa and a standard deviation (controlling the width of the Gaussian function) of 100 hPa.”.

Typos

1. Line 2 in paragraph 2 on page 7. “(Fig. 4b))” to “(Fig. 4b)”. 2. Line 6 in paragraph 2 on page 7. “(Fig. 4a))” to “(Fig. 4a)”. 3. Line 1 in paragraph 4 on page 7. “Figure. 4c)” to “Figure 4c”. 4. Line 10 in paragraph 4 on page 7. “(Fig. 4d))” to “(Fig. 4d)”.

Thank you for pointing these out. They have been corrected.

Reviewer 2

This study looks at the contributions of African dust aerosols and sulfate aerosols to tropical Atlantic SST, and hence hurricane activity, since the 1970s. The topic in general has received a lot of attention in the past 10-15 years. The main idea of the current study is the proposed dominance of dust forcing on tropical North Atlantic SST, combined with the hypothesized connection between sulfate and African dust aerosol contributions, through sulfate-forced changes in Sahel rainfall/winds. However, these links are not demonstrated convincingly. The result is a manuscript in which all of the main pieces (importance of sulfate and dust aerosol forcing on Atlantic SST, importance of multidecadal variations of SST for hurricane activity) have already been shown in previous papers. Significant additional work would be needed to test the proposed ideas and come up with novel and robust results.

Major comment 1

GPI discussion: Observed TC activity was 80% lower in 1970-90 compared to 1990-2010, yet even a very large increase in dust optical depth of 0.3 would result in only a 55% change in TC genesis. So the observed change in DAOD (and resultant change in SST) contributed only about 10-15% of the TC genesis reduction. Other factors like shear and vorticity must be important, and its unclear if they are related to dust-induced changes in SST.

The activity metric presented is the power dissipation index, and we would only expect its variation to be the same as GPI if the duration and intensity of the storms did not change during the hurricane drought. In reality, as we mention in the introduction, the largest fractional decrease occurred for major hurricanes, so we expect that the average storm intensity decreases in addition to the storm number. Finally, since environmental conditions were less favorable in the 1970s and 1980s, storm duration also decreases, which also has an impact on PDI. So a large PDI decrease can occur in conjunction to a much more modest GPI decrease.

To clarify this, we have added the following paragraph to the introduction: “Separating PDI into its three component, storm number, intensity and duration (see methods), shows that the 55% change in PDI is due to a 22% change in hurricane number, a 14% change in intensity (proportional to wind speed cubed) and an 11% change in storm duration. In other words, during the 1970s and 1980s hurricane drought, storms were less numerous (the biggest contribution to PDI change), weaker and shorter-lived.”.

Note that, as stated in the *General response* section, the reference period is now 1960-2017 so the results are numerically different from the previous version (e.g., 55% PDI change by comparison to 80% in the previous version). We also expanded the PDI subsection of the Methods: “It can be shown (Emanuel, 2007) that PDI during a given period can be represented as the product between the number of storms during that period N , the intensity-weighted duration of the storms D , and the averaged intensity of the storms I . Hence, $PDI = N \times D \times I$, where

$$D = \frac{1}{N} \sum_{i=1}^N D_i, \quad D_i = \frac{\int_0^{\tau} V_{max} dt}{V_{max}}, \quad I = \frac{1}{ND} \sum_{i=1}^N \int_0^{\tau} V_{max}^3 d\tau,$$

and V_{max} is the lifetime maximum intensity. When the variations of I , N and D between two averaging periods are small enough, we can approximate fractional variations in PDI as the sum of the fractional variations of I , N and D , so that

$$\frac{\Delta PDI}{PDI} \approx \frac{\Delta N}{N} + \frac{\Delta D}{D} + \frac{\Delta I}{I}.$$

”

Major comment 2

More details are needed about how the surface radiation from the single column model forces the ocean. Is there any coupling between the atmosphere and ocean (damping of SST through surface turbulent heat fluxes)? Or is this done through damping to climatological SST (or something else)? A 2-m deep slab mixed layer will be extremely sensitive to changes in the surface radiative heat flux. How does your dust-forced SST anomaly vary with changes in mixed layer depth? Fig. 3: Why are SSTs 30-31C in (a)? That’s unrealistically high.

We agree with the reviewer that all of these points should be clarified. Sea-surface temperature at equilibrium in this model is entirely determined by the balance between net radiation and turbulent enthalpy fluxes at the surface, with possible effects of ocean heat flux convergence set to zero. For that reason, the value of SST at equilibrium does not depend on mixed layer depth (MLD). The time needed to reach equilibrium does depend on MLD, and to make the simulations computationally cheap, we initially chose an arbitrarily small MLD (2m). This being said, these simulations are not computationally costly anyway and we agree that this is a confusing choice for readers. In addition, the time taken to reach equilibrium does matter. Hence, for the tropical North-Atlantic summer, we choose a 25 m MLD based on Evan et al. (2012) and de Boyer Montégut et al. (2004). We note that, with this choice, the sensitivity of SST to optical depth at equilibrium in the updated figure is essentially the same as in the previous version of the paper ($\approx 1.4 \text{ K } \tau^{-1}$). With a mixed layer depth of 25 m, after 1 month, the SST perturbation has reached 75% of its equilibrium value and after 2 months, the perturbation has reached over 95% of its equilibrium value. This is much faster than the time it would take to reach equilibrium in an RCE simulation, because of the WTG constraint on the atmosphere. Hence, SST perturbations can essentially reach equilibrium within a dust season. Finally, the temperature of the ocean was a bit high because the albedo of the surface in the simulations was a bit low. We increased it to 0.065, which is reasonable, and now the temperature at $\tau = 0$ is 27.5 degrees C.

To clarify these points, we have added the following description at the end of the methods paragraph for the MIT SCM: “In the simulations, the ocean is represented by a slab, the temperature of which varies in response to changes in the balance between net radiation and turbulent enthalpy fluxes at the surface. Possible effects of ocean heat flux convergence are set to zero. We choose a 25 m slab thickness, which is representative of summer mixed layers depths (de Boyer Montégut et al., 2004; Evan et al., 2012). This choice has no influence on equilibrium perturbation SST, but only on the time it takes to reach this equilibrium. In our WTG-constrained simulations, the SST perturbation reaches 75% of its equilibrium value after a month, and over 95% of its equilibrium value after two months (not shown), much faster than radiative convective equilibrium simulations.”

Major comment 3

It's hard to believe that increased dust radiative forcing, which it is claimed was ultimately driven by higher levels of sulfate aerosols (through their impact on SST and atmospheric circulation and Sahel precipitation/winds), could be the main driver of lower SST during the 1970s and 80s. You're saying that a 0.03C decrease in SST due to sulfate aerosols caused a 0.1C decrease in SST due to dust, if I understand correctly. That would mean that Sahel rain/winds/dust are extremely sensitive to tropical Atlantic SST, which has not been shown. That is the weakest part of the argument. Since you are claiming that almost all of the negative SST anomaly during the 70s and 80s was caused by sulfate and dust aerosols, and the positive dust anomalies were driven by sulfate forcing, the impact of sulfate aerosols (and/or tropical Atlantic SST) on dust optical depth needs to be quantified convincingly. Based on the current arguments in the manuscript, it seems more likely that the dust radiative forcing has been overestimated, or natural variability in the ocean (AMOC, NAO, etc.) was an important driver of the cooler SSTs during the 70s and 80s (or there is a natural multidecadal cycle of dust emission (unrelated to SST forcing) that dominates the radiative forcing of tropical North Atlantic SST).

We agree that this point need to be clarified but we disagree that our arguments imply that a 0.03 K SST decrease forced by sulfates is the cause of an additional 0.1 K decrease through a dust feedback. First, European sulfates can cause drought in the Sahel, even with fixed SST, by weakening directly the Saharan Heat low and the West African Monsoon (Dong et al., 2014). This needed to be clarified and we have amended the paper accordingly. Second, due to scavenging by dust aerosols, it is likely that the sulfate radiative forcing was smaller directly under the dust plume (Li et al., 1996; Li-Jones and Prospero, 1998), and thus in the main development region, than it was in other regions of the Atlantic basin. Hence, we do not think the temperature depression in the main development region itself has a large influence on Sahel drought and dust emissions. To clarify this, we have updated the paragraph describing the mechanism in the introduction, as well as amending the paper elsewhere.

“At the peak of SO₂ emissions, in the 1970s and 1980s, sulfate aerosols originating from Europe were swept southward across the Mediterranean and over North Africa by dominant lower tropospheric winds (Lelieveld et al., 2002), where they acted to cool the regional climate (Marmer et al., 2007), which weakened the Saharan heat low and the West-African monsoon (Dong et al., 2014). Simultaneously, sulfate aerosol forcing around the North-Atlantic basin, due mostly to emissions from Europe and North-America weakened the inter-hemispheric temperature gradient during summer (Held et al., 2005), which may have reduced the northward extent of the African monsoon. Both mechanisms have been shown by multiple studies to have caused or worsened a concurrent drought in the Sahel (Biasutti, 2011; Allen et al., 2015; Ackerley et al., 2011; Westervelt et al., 2018; Zhang et al., 2022).”

Major comment 4

It's already been shown that African dust has contributed to decadal-multidecadal variations of tropical Atlantic SST and hurricane activity (Evan et al., Nat. Geosci., 2011; Wang et al., J. Clim., 2012), so that is not new. It's also known that sulfate aerosols have had a big impact on tropical Atlantic climate, and it's been argued that we are currently in a negative phase of the AMO due to anthropogenic aerosol and GHG forcing (Mann et al., GRL, 2014). The new idea in the present study seems to be the connection/interaction between sulfate and dust aerosols. However, more robust connections between aerosols, SST, and hurricane activity need to be established in order to make the present study novel and convincing enough to be published.

We agree that much of what we reported is already known to different subfields of the geosciences. Here we are putting these previous works together as a sequence that can explain both the 20th century dust variations and the fact that the hurricane drought is poorly represented in climate models. From everything that we understand, the hypothesis presented here is a plausible one and a valuable contribution to the scientific discourse on the topic. More details are available in the *General response* section at the beginning of this document. We clarify these points in a new paragraph in the discussion section.

“Individually, the results presented here do not depart much from existing literature. In different sub-fields of climate science, it is fairly well accepted that there was a hurricane drought in the 1970s and 1980s (Nyberg et al., 2007), that hurricane activity correlates well with temperature anomalies (Emanuel, 2007), that Saharan dust variability can explain an important fraction of SST variability (Evan et al., 2012; Strong et al., 2015) and hurricane activity variability (Strong et al., 2018) at long time scales, that dust cover over the tropical North-Atlantic correlate well with dry conditions in the Sahel (Prospero and Lamb, 2003), and that anthropogenic aerosols were an important contributor to drought in the Sahel in the 1970s and 1980s (Ackerley et al., 2011). However, a mechanism by which anthropogenic sulfate aerosol forcing leads to enhanced dust emissions and a decrease in hurricane activity has, to our knowledge, never been proposed. Such a mechanism, although difficult to verify using model simulations, would help answer outstanding scientific questions regarding hurricane activity past and future, and have important implications that warrant further discussion by the community.”

To support the value of this paper, we would respectfully like to bring the reviewer's attention to their statement that “[...] it's been argued that we are currently in a negative phase of the AMO due to anthropogenic aerosols [...]”. The word “Oscillation” in the acronym “AMO”, strongly suggests a internal periodicity which is incompatible with the idea that the observed variations were forced by non-periodic anthropogenic forcing. In a more recent paper, the very author cited to support the reviewer's statement (Michael Mann) shows that

“[...] there is no compelling evidence for internal multidecadal oscillations in the climate system.” (Mann et al., Science, 2021). As the reviewer correctly pointed out, anthropogenic aerosols do explain a large fraction of the observed North-Atlantic basin variability, but there remains a lot of confusion regarding possible roles for natural variability, including for dust variations, which are often presented as due to multidecadal internal variability. We think that our paper would help to clarify these roles and disentangle the concept of AMO from the role of anthropogenic aerosols.

Reviewer 3

This paper explores the role of dust feedbacks in driving the amplitude of MDR SSTs, and hence Atlantic hurricane activity, during the 1970s-1980s. The authors suggest that anthropogenic aerosol emissions drove a reduction in Sahel precipitation, which then led to a decrease in dust transport across the tropical Atlantic and that it is this feedback that was responsible for the majority (77%) of the observed MDR SST cooling during this period. This is an interesting paper, and an important topic, but I think the authors need to provide a clearer narrative about the novelty of their results, a clearer explanation of methods used and finally propagate the uncertainties in their analysis. Hence I recommend major revisions in the hope that these issues can be addressed.

I think it is well known that decreases in Sahel precip during the 1970s-1980s occurred coincident with an increase in dust emissions from the region and that this increases the dust optical depth across the MDR region of the tropical North Atlantic and that this *potentially* offers a dust feedback onto MDR SSTs and hence North Atlantic tropical cyclone activity. In particular, I am aware that Amato Evan has written a number of papers on the topic over the years (which you cite). So I think the particular novelty of this paper needs to be clearly framed in this context.

We agree that the novelty of the paper needs to be better framed, a point that was also raised by the other reviewers (see the *General response* section at the beginning of this document). To clarify this, we have added to the discussion:

“Individually, the results presented here do not depart much from existing literature. In different sub-fields of climate science, it is fairly well accepted that there was a hurricane drought in the 1970s and 1980s (Nyberg et al., 2007), that hurricane activity correlates well with temperature anomalies (Emanuel, 2007), that Saharan dust variability can explain an important fraction of SST variability (Evan et al., 2012; Strong et al., 2015) and hurricane activity variability (Strong et al., 2018) at long time scales, that dust cover over the tropical North-Atlantic correlate well with dry conditions in the Sahel (Prospero and Lamb, 2003),

and that anthropogenic aerosols were an important contributor to drought in the Sahel in the 1970s and 1980s (Ackerley et al., 2011). However, a mechanism by which anthropogenic sulfate aerosol forcing leads to enhanced dust emissions and a decrease in hurricane activity has, to our knowledge, never been proposed. Such a mechanism, although difficult to verify using model simulations, would help answer a number of outstanding scientific questions and have important implications that warrant further discussion by the community.”

This paper attempts to quantify the strength of the dust feedback and hence examine the extent to which dust feedbacks are necessary in order to realise the amplitude of MDR SST cooling during the 1970s-1980s. However, there are many other mechanisms that could explain the MDR SST, for example the circulation changes linked to reduced Sahel rainfall (i.e. the African monsoon/ITCZ) may also change the low-level winds over the MDR that then modify the SSTs via a wind-evaporation-cloud mechanism – this mechanism might also not be well captured by present climate models. Furthermore, the direct aerosol forcing mechanism might itself not be well captured by current climate models – there is large uncertainty in regional aerosol forcing and the simulation of aerosol-cloud indirect effects. Finally, there is quite a lot of evidence to suggest that models may underrepresent the strength of the connection between the North Atlantic sub-polar gyre and tropical SSTs – the AMV horseshoe does not extend far enough south (or weakens too much). Whilst I am not saying that the authors need to go into any detail on these alternative hypotheses, I think it does make it essential to outline the uncertainties in their analysis so readers can make their own assessment on the confidence in these results – something that I feel is missing at the moment.

We agree that the wind and cloud feedbacks need to be better addressed in the paper. Hence we have added caveats to the paragraphs where we introduce and later discuss the dust feedback. They are reproduced below:

In the introduction: “Further we propose that the radiative effects of this dust feedback might have exceeded those of the sulfates themselves in the tropical North Atlantic main development region. An important caveat to this statement is that the radiative effects of the sulfate aerosols themselves are uncertain (Zhang et al., 2013). Perturbations to SST in the tropical North Atlantic are also thought to be enhanced by surface wind speed and cloud feedbacks, which may be related to the effects of dust (Evan et al., 2011; Doherty and Evan, 2014).”

In the discussion: “The perspective presented here suggests that we shouldn’t expect the currently high hurricane activity to decrease in the upcoming decades in association with a return to the negative phase of a natural oscillation. We do not, however, exclude an important role of natural variability at quasi-decadal or shorter time scales. An additional important caveat is that the proposed mechanism does not account for wind and cloud feedbacks, which may also influence the amplitude of the tropical North-Atlantic variability at

long time scales (Bellomo et al., 2016).”

While we agree with some of the caveats raised by the reviewer, we do not think that the fact that the AMV horseshoe does not extend far enough south in models should be viewed as an alternative mechanism to the role of dust. On the contrary, we believe that this precise shortcoming of the climate models can be best explained by our proposed dust feedback mechanism.

The key result, upon which the paper conclusions are based, is that the dust optical depth change in the 1970s-1980s is estimated to be 0.07 which then is then calculated to lead to a reduction in MDR SST of 0.1 K. But I do not think the method to arrive at this is fully explained and the uncertainties are not propagated. It is not clear to me how the Barbados optical depth timeseries in Fig. 2a is arrived at. The Methods section also seems to lack detail – I think the satellite measurements need to be shown in some way to see how they are used to ‘rescale’ the Barbados measurements and hence arrive at the 0.07 optical depth change and crucially what the estimated confidence interval is on this estimate. Then the single-column model simulations could also be better explained, including key assumptions and limitations. There is a brief discussion of the impact of modifying the dust optical properties, but this sensitivity is not followed through to an uncertainty in the 0.1K, or the final 77% figure. So essentially, there should be confidence intervals on the optical depth, the sensitivity of SST to SW, and hence the SST reduction and fraction of variance explained. I think if these are propagated through then there would be quite a large uncertainty associated with the impact of dust feedbacks. I am not saying that this in anyway invalidates the results presented, but it might have implications for the conclusions drawn by the reader when comparing it to other evidence.

We agree that the method needs to be clarified so that readers can interpret the results. We also agree that there are substantial uncertainties at every step of this process, and we attempt to express them better here. We updated the method describing the dust estimates with the following paragraph:

“First we estimate dust optical depth over the main development region from 1965 to 2012. To do so, because the AVHRR record begins in 1982 and because dust concentration peaks in the early 1980s, we rescale the Barbados dust concentration timeseries so that dust optical depth in the 5-years period from 1982 to 1987 in the rescaled timeseries is the same as in the AVHRR dust optical depth peak averaged over the main development region for that period. This method, which assumes that zero dust concentration at Barbados implies zero dust optical depth over the main development region, is similar to the method used by Strong et al. (Strong et al., 2015, 2018). A caveat is that the AVHRR data exhibits substantial uncertainties and appears to underestimate optical depths in the main development region by comparison to the more recent MODIS satellite products (Voss and Evan, 2020). We cannot

use MODIS data for this rescaling because it does not extend sufficiently far back in time, but only to 2001. Hence, this rescaling yields a coarse estimate of how dust optical depth may have varied over the main development region since the 1960s, a time period which isn't covered by the satellite measurements. Then, similarly to Mahowald et al. (2010) (Mahowald et al., 2010), to further extend the dust record in time, we compute a regression of Barbados dust on a Sahel precipitation index (mm/month) extending from 1901 to 2017. Using this method, we estimate that the average dust optical depth over the main development region in the 1970s and 1980s was about 0.230, and that the average dust cover over the 1960-2017 period where we can reasonably trust hurricane observations was about 0.187 so that dust optical depth during the hurricane drought was about 0.043 higher on average. To assess the uncertainty on this estimate, we compute a bias-corrected and accelerated (BCa) bootstrap confidence interval on the regression coefficient between Sahel precipitation index and the Barbados-based dust optical depth record using 1000 data samples and obtain an interval of 0.043 ± 0.010 . Using the Wald method independently yields a very similar estimate of the confidence interval. Since we do not explicitly compute an estimate of the uncertainty on the Barbados-based optical depth record, this uncertainty is likely an underestimate. We also note that these estimates are sensitive to the definition of the main development region.”

We have expanded the single-column model section of the methodology and have added a sentence to describe the most important limitation of these simulations in our opinion. The sentence is “We note that while our simulations parameterize the effects of gravity waves radiating local temperature perturbations away - an important constraint by the large scale environment on the response to a local forcing - they do not represent other potentially important environmental responses to a local forcing such as possible changes in near surface winds.”.

Finally, we have added an estimate of bounds for the SST depression associated with dust in the 1970s and 1980s, including the over-reflective and over-absorptive dust simulations. The paragraph is: “ Modifying dust optical properties such as the single scattering albedo or the asymmetry parameter to unlikely values yield sensitivities between $\delta SST/\delta\tau = -1.3 \text{ K } \tau^{-1}$ and $\delta SST/\delta\tau = -1.8 \text{ K } \tau^{-1}$ (gray symbols). Combining these bounds in dust optical properties with the uncertainty on dust optical depth yields bounds of -0.04 K and -0.10 K on the 1970s and 1980s main development region temperature anomaly that can be explained by dust. Hence, dust can explain a substantial fraction of the observed -0.13 K anomaly during that period, but not all of it. The remainder may be due to direct forcing by sulfates (e.g., Booth et al., 2012) and to wind and cloud feedbacks (e.g., Bellomo et al., 2016). We note that certain uncertainties, like that on the satellite retrievals of dust optical depth (Voss and Evan, 2020) are hard to assess and were not accounted for in these bounds.”

I also wonder if there other lines of evidence that the authors could use to add weight to their analysis. For example, does natural interannual variability in Sahel precip driven by

ENSO give the opportunity to validate some of the dust optical properties and SST sensitivity using more modern satellite observations, or ground data? Or, does the spatial pattern of dust induced SST variability provide a key fingerprint?

We thank the reviewer for this advice, but given the interdisciplinary nature of this paper, we think it would lead to a much longer paper. For the sake of brevity, we prefer to keep this for future work.

Part of the motivation appears to be that present climate models do not simulate the amplitude of MDR SST cooling in the 1970s-1980s, but the authors do not provide direct evidence of this, or even quantify this from the cited literature and furthermore I'm not sure the references cited show this clearly. The authors suggest that models 'account for sulfate radiative forcing' but we do not know the extent to which they model the processes correctly.

We agree with the fact that model representation of aerosols is uncertain and have added a sentence to outline that caveat. We also agree that better references could be chosen to discuss the amplitude of the MDR cooling in models. We have introduced a few more:

“However, despite accounting for sulfate radiative forcing, climate model simulations fail to represent the magnitude of SST variations at multidecadal time scales, in the tropical North-Atlantic and the main development region (Bellomo et al., 2018; Voltaire et al., 2019; Watanabe and Tatebe, 2019; Vågane, 2020). As a result, models also fail to capture the full observed depression of hurricane activity of the 1970s and 1980s (Dunstone et al., 2013; Villarini and Vecchi, 2013).”

We note that these references, which analyse both CMIP5 and CMIP6 models, do not present the underestimated SST depression in the MDR as one of their key results, but they do mention it as a part of their work. We will add that, while they may exist we have not yet found a modelling paper which claims to capture the magnitude of the low-frequency MDR SST variability.

I don't think the role of anthropogenic aerosols in driving Sahelian dust variability is well described. As I see it there are two possible mechanisms: i) that European and North American anthropogenic aerosol emissions (mainly sulfate) drove interhemispheric temperature gradients that drove meridional shifts in the ITCZ, including over the Sahel, leading to the reduce Sahel precip. ii) that European aerosols alone are advected south, reduce surface SW heating over North Africa directly weakening the Saharan heat low and the West African Monsoon resulting in a decrease in Sahel precipitation. From the paper I'm not sure which of these mechanisms the authors are appealing to? See Dong et al 2014 for the latter in an atmosphere only climate model experiment (<https://journals.ametsoc.org/view/journals/clim/27/18/jcli-d-13-00769.1.xml>)

We thank the reviewer for pointing this out. This is indeed not something that was properly clarified in the paper. We do not think that the two mechanisms are mutually exclusive as both have received support from simulations, and we clarified this in the following paragraph:

“At the peak of SO₂ emissions, in the 1970s and 1980s, sulfate aerosols originating from Europe were swept southward across the Mediterranean and over North Africa by dominant lower tropospheric winds (Lelieveld et al., 2002), where they acted to cool the regional climate (Marmer et al., 2007), which weakened the Saharan heat low and the West-African monsoon (Dong et al., 2014). Simultaneously, sulfate aerosol forcing around the North-Atlantic basin, due mostly to emissions from Europe and North-America weakened the inter-hemispheric temperature gradient during summer (Held et al., 2005), which may have reduced the northward extent of the African monsoon. Both mechanisms have been shown by multiple studies to have caused or worsened a concurrent drought in the Sahel (Biasutti, 2011; Allen et al., 2015; Ackerley et al., 2011; Westervelt et al., 2018; Zhang et al., 2022).”

References

- Ackerley, D., Booth, B. B., Knight, S. H., Highwood, E. J., Frame, D. J., Allen, M. R., and Rowell, D. P. (2011). Sensitivity of twentieth-century sahel rainfall to sulfate aerosol and co2 forcing. *Journal of Climate*, 24(19):4999–5014.
- Allen, R. J., Evan, A. T., and Booth, B. B. (2015). Interhemispheric aerosol radiative forcing and tropical precipitation shifts during the late twentieth century. *Journal of Climate*, 28(20):8219–8246.
- Andrews, M. B., Ridley, J. K., Wood, R. A., Andrews, T., Blockley, E. W., Booth, B., Burke, E., Dittus, A. J., Florek, P., Gray, L. J., et al. (2020). Historical simulations with hadgem3-gc3. 1 for cmip6. *Journal of Advances in Modeling Earth Systems*, 12(6):e2019MS001995.
- Bellomo, K., Clement, A. C., Murphy, L. N., Polvani, L. M., and Cane, M. A. (2016). New observational evidence for a positive cloud feedback that amplifies the atlantic multidecadal oscillation. *Geophysical Research Letters*, 43(18):9852–9859.
- Bellomo, K., Murphy, L. N., Cane, M. A., Clement, A. C., and Polvani, L. M. (2018). Historical forcings as main drivers of the atlantic multidecadal variability in the cesm large ensemble. *Climate Dynamics*, 50(9):3687–3698.
- Biasutti, M. (2011). Atmospheric science: A man-made drought. *Nature Climate Change*, 1(4):197.
- Booth, B. B., Dunstone, N. J., Halloran, P. R., Andrews, T., and Bellouin, N. (2012). Aerosols implicated as a prime driver of twentieth-century north atlantic climate variability. *Nature*, 484(7393):228.

- Boucher, O., Servonnat, J., Albright, A. L., Aumont, O., Balkanski, Y., Bastrikov, V., Bekki, S., Bonnet, R., Bony, S., Bopp, L., et al. (2020). Presentation and evaluation of the ipsl-cm6a-lr climate model. *Journal of Advances in Modeling Earth Systems*, 12(7):e2019MS002010.
- de Boyer Montégut, C., Madec, G., Fischer, A. S., Lazar, A., and Iudicone, D. (2004). Mixed layer depth over the global ocean: An examination of profile data and a profile-based climatology. *Journal of Geophysical Research: Oceans*, 109(C12).
- Doherty, O. M. and Evan, A. T. (2014). Identification of a new dust-stratocumulus indirect effect over the tropical north atlantic. *Geophysical Research Letters*, 41(19):6935–6942.
- Dong, B., Sutton, R. T., Highwood, E., and Wilcox, L. (2014). The impacts of european and asian anthropogenic sulfur dioxide emissions on sahel rainfall. *Journal of Climate*, 27(18):7000–7017.
- Dunstone, N., Smith, D., Booth, B., Hermanson, L., and Eade, R. (2013). Anthropogenic aerosol forcing of atlantic tropical storms. *Nature Geoscience*, 6(7):534.
- Emanuel, K. (2007). Environmental factors affecting tropical cyclone power dissipation. *Journal of Climate*, 20(22):5497–5509.
- Emanuel, K. and Sobel, A. (2013). Response of tropical sea surface temperature, precipitation, and tropical cyclone-related variables to changes in global and local forcing. *Journal of Advances in Modeling Earth Systems*, 5(2):447–458.
- Evan, A. T., Foltz, G. R., and Zhang, D. (2012). Physical response of the tropical–subtropical north atlantic ocean to decadal–multidecadal forcing by african dust. *Journal of climate*, 25(17):5817–5829.
- Evan, A. T., Foltz, G. R., Zhang, D., and Vimont, D. J. (2011). Influence of african dust on ocean–atmosphere variability in the tropical atlantic. *Nature Geoscience*, 4(11):762.
- Held, I., Delworth, T., Lu, J., Findell, K. u., and Knutson, T. (2005). Simulation of sahel drought in the 20th and 21st centuries. *Proceedings of the National Academy of Sciences*, 102(50):17891–17896.
- Lelieveld, J., Berresheim, H., Borrmann, S., Crutzen, P., Dentener, F., Fischer, H., Feichter, J., Flatau, P., Heland, J., Holzinger, R., et al. (2002). Global air pollution crossroads over the mediterranean. *Science*, 298(5594):794–799.
- Li, X., Maring, H., Savoie, D., Voss, K., and Prospero, J. (1996). Dominance of mineral dust in aerosol light-scattering in the north atlantic trade winds. *Nature*, 380(6573):416.

- Li-Jones, X. and Prospero, J. (1998). Variations in the size distribution of non-sea-salt sulfate aerosol in the marine boundary layer at barbados: Impact of african dust. *Journal of Geophysical Research: Atmospheres*, 103(D13):16073–16084.
- Mahowald, N. M., Kloster, S., Engelstaedter, S., Moore, J. K., Mukhopadhyay, S., McConnell, J. R., Albani, S., Doney, S. C., Bhattacharya, A., Curran, M., et al. (2010). Observed 20th century desert dust variability: impact on climate and biogeochemistry. *Atmospheric Chemistry and Physics*, 10(22):10875–10893.
- Marmner, E., Langmann, B., Fagerli, H., and Vestreng, V. (2007). Direct shortwave radiative forcing of sulfate aerosol over europe from 1900 to 2000. *Journal of Geophysical Research: Atmospheres*, 112(D23).
- Nyberg, J., Malmgren, B. A., Winter, A., Jury, M. R., Kilbourne, K. H., and Quinn, T. M. (2007). Low atlantic hurricane activity in the 1970s and 1980s compared to the past 270 years. *Nature*, 447(7145):698–701.
- Prospero, J. M. and Lamb, P. J. (2003). African droughts and dust transport to the caribbean: Climate change implications. *Science*, 302(5647):1024–1027.
- Robson, J., Aksenov, Y., Bracegirdle, T. J., Dimdore-Miles, O., Griffiths, P. T., Grosvenor, D. P., Hodson, D. L., Keeble, J., MacIntosh, C., Megann, A., et al. (2020). The evaluation of the north atlantic climate system in ukesm1 historical simulations for cmip6. *Journal of Advances in Modeling Earth Systems*, 12(9):e2020MS002126.
- Shaevitz, D. A. and Sobel, A. H. (2004). Implementing the weak temperature gradient approximation with full vertical structure. *Monthly weather review*, 132(2):662–669.
- Sobel, A. H. and Bretherton, C. S. (2000). Modeling tropical precipitation in a single column. *Journal of climate*, 13(24):4378–4392.
- Strong, J. D., Vecchi, G. A., and Ginoux, P. (2015). The response of the tropical atlantic and west african climate to saharan dust in a fully coupled gcm. *Journal of Climate*, 28(18):7071–7092.
- Strong, J. D., Vecchi, G. A., and Ginoux, P. (2018). The climatological effect of saharan dust on global tropical cyclones in a fully coupled gcm. *Journal of Geophysical Research: Atmospheres*, 123(10):5538–5559.
- Vågane, J. S. (2020). Atlantic multidecadal variability (amv) in the norwegian earth system model. Master’s thesis, The University of Bergen.
- Villarini, G. and Vecchi, G. A. (2013). Projected increases in north atlantic tropical cyclone intensity from cmip5 models. *Journal of Climate*, 26(10):3231–3240.

- Voldoire, A., Saint-Martin, D., Sénési, S., Decharme, B., Alias, A., Chevallier, M., Colin, J., Guérémy, J.-F., Michou, M., Moine, M.-P., et al. (2019). Evaluation of cmip6 deck experiments with cnrm-cm6-1. *Journal of Advances in Modeling Earth Systems*, 11(7):2177–2213.
- Voss, K. K. and Evan, A. T. (2020). A new satellite-based global climatology of dust aerosol optical depth. *Journal of Applied Meteorology and Climatology*, 59(1):83–102.
- Watanabe, M. and Tatebe, H. (2019). Reconciling roles of sulphate aerosol forcing and internal variability in atlantic multidecadal climate changes. *Climate Dynamics*, 53(7):4651–4665.
- Westervelt, D. M., Conley, A. J., Fiore, A. M., Lamarque, J.-F., Shindell, D. T., Previdi, M., Mascioli, N. R., Faluvegi, G., Correa, G., and Horowitz, L. W. (2018). Connecting regional aerosol emissions reductions to local and remote precipitation responses. *Atmospheric Chemistry and Physics*, 18(16):12461–12475.
- Zhang, R., Delworth, T. L., Sutton, R., Hodson, D. L., Dixon, K. W., Held, I. M., Kushnir, Y., Marshall, J., Ming, Y., Msadek, R., et al. (2013). Have aerosols caused the observed atlantic multidecadal variability? *Journal of the Atmospheric Sciences*, 70(4):1135–1144.
- Zhang, S., Stier, P., Dagan, G., and Wang, M. (2022). Anthropogenic aerosols modulated 20th-century sahel rainfall variability via their impacts on north atlantic sea surface temperature. *Geophysical Research Letters*, 49(1):e2021GL095629.

REVIEWERS' COMMENTS

Reviewer #1 (Remarks to the Author):

My comments on the early version of the paper have been addressed in a satisfactory manner in the revised version. The reviewer does not have any further comments on revised version and the paper is, therefore, acceptable for publication.

Reviewer #2 (Remarks to the Author):

Second review of "Natural and anthropogenic contributions to the hurricane drought of the 1970s-1980s"

The authors have made some improvements, but my opinion of the manuscript has not changed much. There is not enough new material to justify publication. As the authors state, it is already well known that:

1. Anthropogenic aerosols caused cooling in the North Atlantic and drought in the Sahel.
2. Sahel drought (through changes in near-surface winds) increased dust emissions.
3. Increased dust caused cooling of tropical North Atlantic SST and likely generated coupled WES feedback that enhanced SST cooling.
4. Tropical North Atlantic SST is closely related to TC activity.

The proposed novelty is the connection of enhanced sulfate forcing to enhanced dust and decreased hurricane activity. For that to be new and publishable, there would need to be a rigorous quantification of the connections and pathways (mechanisms, regional signals) between anthropogenic aerosol emissions and dust and/or between dust and hurricane activity. I don't see that level of in-depth analysis, just broad statistical relationships.

Reviewer #3 (Remarks to the Author):

I thank the authors for their frank assessment of their work in light of the reviewer comments. I also thank them for expanding on the methods and better quantifying the uncertainties on their analysis as suggested. I believe the paper is now in a state where it could be published in a journal.

The remaining question I have still relates to the novelty of the results presented in the paper and in particular whether the paper provides a sufficient advance to warrant publication in Nature Communications. I agree with the authors that there is certainly merit in combining results from different fields, to provide a clearer explanation for how anthropogenic aerosols have impacted North Atlantic tropical storm activity. I think this must ultimately be an editorial decision.

Response to Reviewers: “Natural and Anthropogenic Contributions to the Hurricane Drought of the 1970s-1980s”

Raphaël Rousseau-Rizzi and Kerry Emanuel

General response

We thank the reviewers for their second round of insightful and valuable comments, and for their work in reviewing this paper a second time.

Reviewer 1

My comments on the early version of the paper have been addressed in a satisfactorily manner in the revised version. The reviewer does not have any further comments on revised version and the paper is, therefore, acceptable for publication.

We thank reviewer 1 for the comments, which have greatly improved the paper.

Reviewer 2

Second review of “Natural and anthropogenic contributions to the hurricane drought of the 1970s-1980s”. The authors have made some improvements, but my opinion of the manuscript has not changed much. There is not enough new material to justify publication. As the authors state, it is already well known that:

1. Anthropogenic aerosols caused cooling in the North Atlantic and drought in the Sahel.
2. Sahel drought (through changes in near-surface winds) increased dust emissions.
3. Increased dust caused cooling of tropical North Atlantic SST and likely generated coupled WES feedback that enhanced SST cooling.
4. Tropical North Atlantic SST is closely related to TC activity.

The proposed novelty is the connection of enhanced sulfate forcing to enhanced dust and decreased hurricane activity. For that to be new and publishable, there would need to be a rigorous quantification of the connections and pathways (mechanisms, regional signals) between anthropogenic aerosol emissions and dust and/or between dust and hurricane activity. I don't see that level of in-depth analysis, just broad statistical relationships.

We agree with the individual statements 1 to 4 presented by the reviewer. However, we think that by collecting these agreed-upon facts, and by collecting literature on conventionally separate topics, we have managed to propose a novel pathway for anthropogenic aerosols to influence tropical SSTs, which is consistent simultaneously with current gaps in the scientific literature, like why models fail to capture the hurricane drought magnitude, and with the evidence presented in our paper. We do also acknowledge clearly in the paper that verifying this mechanism with high confidence is difficult, but we hope that this publication will stir the debate on this question, and lead to further progress on our understanding of the hurricane drought.

Reviewer 3

I thank the authors for their frank assessment of their work in light of the reviewer comments. I also thank them for expanding on the methods and better quantifying the uncertainties on their analysis as suggested. I believe the paper is now in a state where it could be published in a journal.

The remaining question I have still relates to the novelty of the results presented in the paper and in particular whether the paper provides a sufficient advance to warrant publication in Nature Communications. I agree with the authors that there is certainly merit in combining results from different fields, to provide a clearer explanation for how anthropogenic aerosols have impacted North Atlantic tropical storm activity. I think this must ultimately be an editorial decision.

We thank reviewer 3 again for the comments, which have greatly improved the paper.